

# On the need of a time and location dependent estimation of the NDSI threshold value for reducing existing uncertainties in snow cover maps at different scales

Stefan Härer[1], Matthias Bernhardt[1], Matthias Siebers[2], Karsten Schulz[1]

[1]Institute for Water Management, Hydrology and Hydraulic Engineering (IWHW), University for life sciences (BOKU), 1190 Vienna, Austria
[2]Commision for Glaciology, Bavarian Academy of Sciences and Humanities, 80539 Munich, Germany

*Correspondence to*: Matthias Bernhardt (matthias.bernhardt@boku.ac.at)

**Abstract.** Knowledge about the current snow cover extent is essential for characterising energy and moisture fluxes at the earth surface. The snow-covered area (SCA) is often estimated by using optical satellite information in combination with the normalized-difference snow index (NDSI) .The NDSI thereby uses a threshold for the definition if a satellite pixel is assumed to be snow covered or snow free. The spatio-temporal representativeness of the standard threshold of 0.4 is however

questionable at the local scale. Here, we use local snow cover maps derived from ground-based photography to continuously calibrate the NDSI threshold values ($NDSI_{thr}$) of Landsat satellite images at two European mountain sites of the period from 2010 to 2015. Both sites, the Research Catchment Zugspitzplatt (RCZ, Germany) and the Vernagtferner area (VF, Austria), are located within a single Landsat scene. Nevertheless, the long-term analysis of the $NDSI_{thr}$ demonstrated that the $NDSI_{thr}$ at these sites are not correlated and different to the standard threshold of 0.4. For further comparison, a dynamic and locally

optimized NDSI threshold was used as well as another literature threshold value. It was shown that large uncertainties in the prediction of the SCA of up to 24.1% exist in satellite snow cover maps in case the standard threshold of 0.4 is used, but a newly developed calibrated quadratic polynomial model which is accounting for seasonal threshold dynamics can reduce this error. The model  minimizes the SCA uncertainties at the calibration site VF by 50% in the evaluation period and was also able to improve the results at RCZ in a significant way. Additionally, a scaling experiment has shown that the positive effect

of a locally adapted threshold diminishes from a pixel size of 500m and more which underlines the general applicability of the standard threshold at larger scales.

## 1 Introduction

Numerous studies ranging from the local to the global scale have underlined the influence of snow cover on e.g. air temperature, runoff generation, soil temperature and soil moisture (Bernhardt et al., 2012; Deb et al., 2015; Dutra et al., 2012; Dyurgerov,

2003; Liston, 2004; Mankin and Diffenbaugh, 2015; Santini and di Paola, 2015; Tennant et al., 2015). Hence, an accurate



estimation of the spatial extent of the snow pack is fundamental for a suite of applications (Pomeroy et al., 2015). The accuracy of weather and climate models heavily depends on this information, as the range of surface temperatures is instantly limited to a maximum of 0° C in existence of snow and the surface albedo becomes typically significantly enhanced (Agosta et al., 2015; Liston, 2004; Rangwala et al., 2010; Takata et al., 2003; Vavrus et al., 2011). From a hydrological point of view, the

formation of a snow pack has a buffering effect and thus often leads to a transfer of precipitation water from the cold to the warm season of the year (Bernhardt et al., 2014; Viviroli et al., 2011). This leads to a support of summer runoff needed e.g. in agriculture or for sanitary water supply, but can also lead to an intensification of flood events e.g. in case of rain on snow events (Viviroli et al., 2011). With this in mind, information on the current snow distribution is elementary for water resources management (Thirel et al., 2013) and weather forecasting model systems (Dee et al. 2011).

Snow cover distribution is often derived from satellite data and then either used as input for operational models (Butt and Bilal, 2011; Dee et al., 2011; Homan et al., 2011; Tekeli et al., 2005) or for the offline evaluation of modelled snow cover (Bernhardt and Schulz, 2010; Warscher et al., 2013) and snow fall patterns (Maussion et al., 2011).

The used snow-cover mapping approaches can be grouped into three categories: manual interpretation, classification-based, and index-based methods. Manual classification as well as classification-based approaches, are often used in local snow cover

mapping studies. Both are out of the scope of this study as a need for expert knowledge and a high time-demand limit their applicability for large time series data. Hence, we focus on the automatic normalized-difference snow index (NDSI) approach. It was developed by Dozier (1989) and is an established method to identify snow cover in optical satellite images. NOAA/NESDIS which is assimilated into ERA/Interim (Dee et al., 2011; Drusch et al., 2004), or the widely used MODIS snow cover products (Hall and Riggs, 2007; Hall et al., 2002) make use of the NDSI.

The NDSI traces back to band rationing techniques (Kyle et al., 1978; Dozier, 1984) related to the NDVI (Rouse et al., 1974; Tucker, 1979) and is based on the physical principle that snow reflection is significantly higher in the visible range of the spectrum than in mid-infrared. The index ranges between -1 and 1 and a differentiation between snow and no snow is based on a NDSI threshold value ($NDSI_{thr}$) which is commonly assumed to be 0.4 (Dozier, 1989; Hall and Riggs, 2007; Sankey et al., 2015). According to Hall et al. (2001) the accuracy for monthly snow detection using the MODIS product with its standard

threshold is about 95% in non-forested and about 85% in forested areas. Accuracies in this range make NDSI based snow cover products well accepted for global scale applications, but uncertainties have to be expected at the local scale (Härer et al. 2016).

In this context, numerous recent studies have questioned the general applicability of a standard $NDSI_{thr}$ in local snow and glacier monitoring. When calibrating the $NDSI_{thr}$ manually or by automated methods against field data for single scenes, large

deviations from the standard value of 0.4 have been observed. The published values range from 0.18 to 0.7 (Burns and Nolin, 2014; Härer et al., 2016; Maher et al., 2012; Racoviteanu et al., 2009; Silverio and Jaquet, 2009; Yin et al., 2013). The wide range of values show the spatio-temporal variability of the $NDSI_{thr}$ and  raise the question for a valid non-subjective method to define this value.



Maher et al. (2012), for example, assumed a spatially calibrated $NDSI_{thr}$ of 0.7 to be constant over time. The comprehensive work of Yin et al. (2013) compared various automatic entropy-based, clustering-based, and spatial threshold methods to adjust the $NDSI_{thr}$ for specific satellite images. The findings of Yin et al. (2013) are based on single-date comparisons at five sites around the world and were undertaken on a regional scale. The clustering-based image segmentation method developed by

Otsu (1979) compared best to the evaluation data sets, which is why the Otsu method is used as comparative data in here.

Härer et al. (2016) have presented a calibration strategy for satellite derived snow cover maps on the basis of local camera systems. The achieved results have shown that $NDSI_{thr}$ can be distinctly different in course of the snow cover period and that there is a need for a temporal adaption of $NDSI_{thr}$ for achieving valid results in view of the local SCA.

The aim of the presented study is to evaluate the variability of $NDSI_{thr}$ in space and time and to test if this variability does

lead to significant uncertainties in the existing snow cover maps. A scaling exercise which has investigated up to which scale a locally adapted threshold can improve the classification results shows the limits of the fixed threshold approach at the local scale.

We use the camera-based calibration approach (Härer et al., 2013) as reference as it has shown its low error margins in comparison to high resolution locally derived 1m resolution snow maps at RCZ (Härer et al. 2016). The results achieved by

this approach are then compared to the automatic segmentation method of Otsu (1979), which has proven to be one of the most performant snow detection methods available today (Yin et al., 2013) and to the standard threshold of 0.4, as well as to a location specific threshold of 0.7 (Maher et al., 2012). The results will reveal the performance of the different approaches and will clarify for which scales a fixed NDSI threshold can be an adequate solution.

## 2 Study Site and Data

The presented study focuses on two mountain sites in the European Alps, the Research Catchment Zugspitzplatt (RCZ) located in Germany (47°40' N/11°00' E; Bernhardt et al., 2015; Weber et al., 2016) and the Vernagtferner (VF) catchment in Austria (46°52' N/10°49' E; Fig. 1a to c; Abermann et al., 2011). RCZ is a partly glaciered headwater catchment with a spatial extent of about 13.1 km². It stretches from 1371 to 2962 m a.s.l. and is mainly built up by limestone. VF is also an alpine headwater basin with a size of 11.5 km² and a glaciated part of about 7.9 km² (Mayr et al., 2013). It ranges from 2642 to 3619 m a.s.l.

and the pending rock is gneiss.

Both sites are equipped with similar single lens reflex camera systems for monitoring wide parts of the catchments starting from May 2011 at RCZ and from August 2010 at VF. The camera locations at the study sites are depicted in Fig. 1a and b and the camera orientations are Southwest at RCZ and West-Northwest at VF. Both investigation areas are located within a single Landsat scene (Fig. 1c) which guarantees for comparable illumination conditions and allows for a direct comparison of the

$NDSI_{thr}$ between both sites.

Overall, 156 Landsat scenes from Landsat 5 TM, 7 ETM+ and 8 OLI were available for the observation period between 18 August 2010 and 31 December 2015. Suitable satellite image-photograph pairs were available at 15 dates for RCZ and VF,



at one date for RCZ and in 32 cases for VF only. The differences stem from the local weather conditions, from the different lengths of the local photograph time series, and from the restriction that a $NDSI_{thr}$ calibration with PRACTISE or the clustering-based image segmentation from Otsu (1979) can only be applied if there is no full snow coverage in the area.

For the photo rectifaction part in our study, digital elevation models (DEM) with a horizontal resolution of 1 m of RCZ and

VF are used, as well as orthophotos with a sub-meter spatial resolution and topographic maps as additional material to ensure an optimal geometric accuracy.

## 3 Methods

Our study investigates the differenes of automatically derived $NDSI_{thr}$ from a) Landsat satellite imagery and b) terrestrial photography with literature values and displays their effects on the resulting snow cover maps.

Radiometrically and geometrically corrected Landsat Level 1 data was used in combination with the cloud and shadow masking software Fmask of Zhu et al. (2015). Any pixel with a cloud probability exceeding 95% in this analysis was excluded with a surrounding buffer of three pixels (Härer et al., 2016). No atmospheric correction is applied to the Landsat data to facilitate a direct comparison to the majority of studies that apply the NDSI for snow cover mapping (Bernhardt and Schulz, 2010; Maussion et al., 2011; Maher et al., 2012; Warscher et al., 2013; Sankey et al., 2015).

The normalized-difference snow index (NDSI) is calculated in accordance to Dozier (1989) by using green (~0.55 µm) and mid-infrared (MIR, ~1.6 µm) reflectance values:

$$NDSI = \frac{\rho_{\text{green}} - \rho_{\text{MIR}}}{\rho_{\text{green}} + \rho_{\text{MIR}}},$$    (1)

NDSI values can range between -1 and 1 and the $NDSI_{thr}$ defines the NDSI value from which on the satellite pixel is assumed as snow covered. We used fixed $NDSI_{thr}$ values and dynamically derived $NDSI_{thr}$ values in course of this. In case of the fixed

values, the standard of 0.4 and a literature value of 0.7 (Maher et al., 2012) were used. For the dynamic approaches, the clustering-based image segmentation approach from Otsu (1979) and a terrestrial camera-based calibration approach of Härer et al. (2016) were applied.

By using Otsu (1979), the $NDSI_{thr}$ is calibrated by maximizing the between-class variance of the two classes snow and no snow:

$$\max_{-1 \leq NDSI_{thr} \leq 1} \{\sigma_O^2\} = \max_{-1 \leq NDSI_{thr} \leq 1} \{P_s(NDSI_{thr}) \, P_{ns}(NDSI_{thr}) \, [\mu_s(NDSI_{thr}) - \mu_{ns}(NDSI_{thr})]\},$$    (2)

where $P_s$ and $P_{ns}$ are the probabilities of the classes snow and no snow with respect to the $NDSI_{thr}$, and $\mu_s$ and $\mu_{ns}$ are the mean values of these two classes. The probability of $P_s$ is thereby calculated as the number of pixels with NDSI values above the $NDSI_{thr}$ divided through the total number of pixels in the image. $P_{ns}$ calculates the absolute difference of $P_s$ to 1.



It has to be mentioned that we restrict the satellite image area used for deriving $NDSI_{thr}$ in accordance to Otsu (1979) to the catchment area of RCZ and VF to allow for a spatio-temporal variable NDSI threshold value within the investigated satellite scenes and to allow for a direct comparison of the locally derived thresholds.

The second dynamic method to calibrate the $NDSI_{thr}$ of the Landsat data for RCZ and VF uses ground-based photographs as

baseline.

The Matlab software PRACTISE (version 2.1; Härer et al., 2013 and 2016) is utilized first to georectify the available terrestrial photographs and secondly to calibrate the $NDSI_{thr}$. For doing so, overlapping areas in the photograph-satellite image pairs are used. For further understanding, Figure 2 gives an overview of the needed input, the internal processing steps and the generated output data of PRACTISE 2.1. The first program part georectifies the photographies and differences between areas with and

without snow. This results in a high resolution photography-based snow cover map (Fig. 2, left column). The second part calibrates the $NDSI_{thr}$ for the satellite scene of interest and uses the achieved value to calculate a NDSI based satellite snow cover map (Fig. 2, right column).

The photo georectification is based on the assumption that the recorded two-dimensional photograph (Fig. 3, blue colour) is geometrically connected to the three-dimensional real world (Fig. 3, black colour). Knowing the camera type, its lens and

sensor system, as well as the camera location and orientation, a georectification becomes possible if a high resolution digital elevation model (DEM) is available as well.

Having this theoretical background in mind, we outline the single processing steps for a photograph and a Landsat 7 scene of VF on 17 November 2011 (Figures 4 a to e, 5 a to c).

Before the PRACTISE program is used, any possible distortion effects of the photograph caused by the camera lens are

removed by utilising the freely available Darktable software (http://www.darktable.org/) and LensFun parameters (http://lensfun.sourceforge.net/). Now that all data is available and ready, the PRACTISE program evaluation can start.

In a first step information about the camera location and orientation is needed for an georectification of the photography. This information is automatically optimised by using ground control points (GCPs, Fig. 4a). The calculated viewpoint and viewing direction are by default used to perform a viewshed analysis (Fig. 4b). The viewshed is needed for an identification of areas

which are visible from the viewpoint and which are not obscured by topographical features or within a user-specified buffer area around the camera. The respective DEM pixels are then projected to the photo plane (Fig. 4c).

Now, the snow classification module is activated to distinguish between snow covered and snow-free DEM pixels (Fig. 4d). Two major procedures are available for classification. A statistical analysis which is using the blue RGB band (Salvatori et al. 2011) and a principal component analysis (PCA) based approach (Härer et al. 2016). The first is used for shadow-free scenes,

the second for scenes with shaded areas. Härer et al. (2013) and (2016) give more insights into the used classification algorithms and their performance as well as on a third manual option if none of the two classification routines can be applied successfully. The used snow cover maps do have less than 5% misclassificied pixels, which was proven by visual inspection. For this example photograph, the snow classification algorithm utilising a principal component analysis (PCA) was selected to account for the shadow-affected areas in the upper left part of the photograph (Fig. 4d, enlarged view in Fig. 4e).



After the photograph rectification and classification, the remote sensing routine of PRACTISE begins with the identification of satellite pixels that spatially overlap with the photograph snow cover map. It also generates a cloud- and shadow-free satellite image by using fmask (Zhu et al., 2015). The needed NDSI map is calculated in accordance to Eq. (1) by PRACTISE (Fig. 5a). If both, the NDSI satellite map and the corresponding high resolution photograph snow cover map were processed, an iterative

calibration of the NDSI threshold value is started to acquire the best agreement between the local scale (photograph) and the large scale (Landsat) snow cover map by maximising the ratio of identically classified pixels to the overall number of photograph-satellite image pixel pairs $n$ (Aronica et al., 2002):

$$F = \frac{(a+d)}{n},$$
(2)

$a$ thereby represents the number of correctly identified snow pixels and $d$ the same for no snow pixels. $F$ is between 0 and 1

and becomes 1 for a perfect agreement between the two images.

Figure 5b shows the resulting satellite snow cover map superimposed on the photograph snow cover map and a Landsat Look image. A cutout is shown for more detail in Fig. 5c.

It has to be mentioned that the glacier retreat between DEM production years (2007, 2010) and analysis period 2010-2015 has resulted in a discrepancy between real world elevations and the available DEMs, especially in the last years of the observation

period. Figure 6 exemplarily depicts the glacier retreat between 2007 and 2010 by superimposing the ice mass loss on an orthophoto of VF from 2010.

This loss in elevation leads to inaccuracies in the georectification results of the photographs. And a test for the photograph of 28 August 2010 applying the DEM of 2007 and 2010 showed that these georectification issues in turn affect the $NDSI_{thr}$ calibration results. For the DEM from 2007, the calibrated $NDSI_{thr}$ is 0.47 while the correct threshold for the up-to-

date DEM from 2010 is 0.52. As a consequence, we limited the analysis to higher elevated and thus colder areas of the catchment where glacier retreat is marginal (areas north-west of the green line in Fig. 5b and Fig. 6).

To ensure that reducing the spatial overlap between photograph snow cover map and NDSI satellite map does not have any negative effect on the calibrated $NDSI_{thr}$, we firstly calibrated the $NDSI_{thr}$ for the three investigated Landsat scenes in 2010 for the complete and the upper area only. Moreover, we calibrated the $NDSI_{thr}$ for the 44 remaining scenes between 2011 and

2015 using the upper area DEM from 2007 and 2010 to test for a $NDSI_{thr}$ sensitivity in the longer time series. For both approaches, the differences between the calibrated $NDSI_{thr}$ never become larger than 0.01. Hence, we assume that our calibration approach of using the higher elevated areas at VF which is incorporated in PRACTISE by excluding a radius of 1800m around the camera from the analysis (green line in Fig. 5b and Fig. 6) is valid for the complete analysed time series between 2010 and 2015.

We did not find a similar effect on the $NDSI_{thr}$ calibration in our tests at RCZ. Hence, there was no need to remove the glacier areas at RCZ from the analysis.





## 4 Results and Discussion

The NDSI thresholds derived by the two dynamic methods are now discussed and related to static thresholds.

The $NDSI_{thr}$ predicted by the Otsu method are densely grouped around 0.4. This is underlined by a mean of 0.36 and a standard deviation of 0.04 at RCZ and a mean of 0.41 with a corresponding standard derivation of 0.04 at VF (Tbl. 1). The statistics do

not include two dates at VF as no separating $NDSI_{thr}$ could be found by using the Otsu method here (squares in Fig. 7a). This stands in contradiction to the real situation as the photographs do show that there was no full snow coverage at the respective dates which would generally allow for an prediction of $NDSI_{thr}$. This shows that the application of the Otsu method is potentially uncertain in nearly fully snow covered situations. Furthermore, a tendency to slightly higher mean $NDSI_{thr}$ at VF and slightly lower thresholds at RCZ could be detected and the very small observed differences to the standard of 0.4

would not underline the need for a location dependent threshold prediction. Additionaly, the weak seasonal dynamics which can be found at VF would also not require a time dependent calculation of the threshold.

The camera-based method leads in general to a more dynamic $NDSI_{thr}$ in time and to a higher systematic difference of $NDSI_{thr}$ between the two sites. The archived 16 $NDSI_{thr}$ at RCZ and 47 $NDSI_{thr}$ at VF are compared in a first step. The presumption of a comparable $NDSI_{thr}$ for both sites could not be confirmed in this case. Significant differences were detected

despite the fact that both sites are high alpine and are located within a single Landsat scene. Moreover, the calibrated $NDSI_{thr}$ were in large parts significantly different to the standard value of 0.4. Figure 7b and Table 1 illustrate the variability and the range of $NDSI_{thr}$ at both sites. The minimum value at RCZ is 0.15 while the maximum value is 0.39. The values at VF are in general on a higher level and range between 0.35 and 0.74. Both sites thus strongly scatter around their catchment-specific mean value (0.28 at RCZ, 0.57 at VF) but show a characteristic development over the year (Fig. 9) which is also detected in

a significantly weaker form for the Otsu method at VF. Independent of the fact that this seasonal dynamic is comparable for both sites using the camera-based method. Fig. 7b highlights that the correlation coefficient between $NDSI_{thr}$ at RCZ and VF is very low when they are compared on a date by date basis (r=0.17). By contrast, a correlation between the Otsu method and the terrestrial camera-based method at VF of -0.56 is found which however cannot be observed at RCZ between the two methods (r=0.10, Fig. 7a and b).

The results of the camera-based methods require a deeper investigation to analyse if such different $NDSI_{thr}$ are justifiable. Despite the strong scatter and the resulting low correlation, the differences in the catchment-specific mean $NDSI_{thr}$ levels seem to be systematic (Tbl. 1). Topographic characteristics could be a possible reason. These are similar with respect to elevation, slope and aspect but different for the pending rock being limestone at RCZ and gneiss at VF. We hence investigated the NDSI reflectance values for the snow-free bare rock areas within each catchment. This is valid for the complete time series

as the steepest almost vertical rock faces in the catchment are snow-free in all used scenes. Figure 8 presents frequency histograms of these NDSI reflectances for five summer dates. Other seasons were excluded due to the increased probability of fractional snow cover in the Landsat pixels. The tests show that the maximum frequencies after smoothing the histogram are stable for these dates for each catchment. The mean maximum frequency is about -0.34 at RCZ and 0.01 at VF. The mean





NDSI reflectance difference of the rocks at RCZ and VF amounts to about 0.34. This difference is comparable to the mean systematic difference of 0.26 found for the mean calibrated $NDSI_{thr}$ at both sites. It is therefore probable that the different rock types and therewith the background radiation triggers the catchment-specific mean $NDSI_{thr}$ levels which in turn supports the idea of adapting $NDSI_{thr}$ locally.

Next, the effect of the calibrated $NDSI_{thr}$ on the predicted snow covered area (SCA) at RCZ and VF is analysed. The differences between the SCA predicted with the standard threshold of 0.4 and with the Otsu method are in principle small. This can be related to the minor differences between standard $NDSI_{thr}$ and the threshold predicted over Otsu. The absolute differences are 0.05 km² in average for VF and 0.15 km² for RCZ. The effects achieved with the photographic method instead are on a level which questions the applicability of the standard threshold for local investigations. The differences in SCA

inbetween the products using the calibrated $NDSI_{thr}$ and the standard threshold of 0.4 are calculated using the camera-calibrated SCA as baseline which has shown the highest accuracy of the derived snow cover products when compared to the available photoclassifications of PRACTISE (Härer et al. 2016):

$$SCA_{diff\%} = \frac{100\,(SCA_{0.4} - SCA_{cam})}{SCA_{cam}} \tag{3}.$$

The values are between -24.1% at RCZ and +17.2% at VF (Fig. 7c) and reveal how much uncertainty currently exists in NDSI

based snow cover maps on the small scale. The deviations are in general larger at RCZ where the calibrated NDSI threshold values are mainly below 0.4. This means that the SCA is systematically underestimated when using the standard of 0.4. The lower error in percents at VF compared to the error percentages at RCZ can be related to the generally higher snow covered area in the VF catchment. These relative differences result in turn in significantly different absolute SCA  (standard threshold versus calibrated threshold). Here, the highest differences are 1.09 km² at RCZ and 1.67 km² at VF. This is a relevant error

margin especially if the small catchment sizes of only 13.1 km² (RCZ) and 11.5 km² (VF) are taken into account.

Given this finding and the large variability observed in calibrated $NDSI_{thr}$ it is obvious that widely used methods (e.g. Maher et al., 2012) which locally calibrate the $NDSI_{thr}$ for a single date and then apply this threshold at multiple dates are also no solution and can even deteriorate the accuracy compared to the standard threshold method. An example is the application of a calibrated threshold of 0.7 at VF to the complete time series in this catchment. This results in a mean absolute error in SCA of

1.26 km² compared to an average deviation of 0.41 km² for the standard threshold method.

An alternative to the temporally constant threshold methods is a statistical modelling approach fitted to the calibrated $NDSI_{thr}$. This however requires a solid set of calibration data to adjust the model to the observations at multiple dates. VF hence serves as an example for this approach because of its higher data availability. As stated before a seasonal dynamic in the calibrated $NDSI_{thr}$ could be observed at both sites. This temporal development is potentially related to the sun angle, snow age, grain

size or albedo development or other effects. A detailed investigation of the reasons of this effect is beyond this study but will be subject of future studies. A quadratic polynomial model was fitted to the calibrated $NDSI_{thr}$ for the years 2010 to 2013 at VF ($NDSI_{vf}$, Fig. 9). $NDSI_{vf}$ might not exactly reproduce the calibrated thresholds at any time step (r²=0.45; RMSE=0.06)



but the evaluation of this simple model for 2014 and 2015 at VF shows a remarkable reduction in the average SCA error from 0.35 km² when applying the standard threshold of 0.4 down to 0.17 km².

As not any site is equipped with camera infrastructure, it was also tested if the achieved regression model can be transferred to RCZ while including information about the geology dependent offset between the average $NDSI_{thr}$ values. Hence, the model

is fitted to the complete calibrated $NDSI_{thr}$ time series at VF (r²=0.36; RMSE=0.07) and a term for the systematic mean NDSI reflectance difference of the rocks at RCZ and VF is added ($NDSI_{rcz}$, Fig. 8). The evaluation of $NDSI_{rcz}$ seems to slightly underestimate the calibrated $NDSI_{thr}$ at RCZ. Nevertheless, the quadratic polynomial model accounting for the reflectance differences at different sites results in a significant reduction of snow cover mapping uncertainties of 40% as the mean SCA error amounts to 0.18 km² while the application of the standard threshold method causes an average deviation in snow cover

of 0.31 km² in RCZ. Given the assumption that the seasonal dynamic and the correction factor are generally applicable, the presented seasonal model derived from the multi-year use of PRACTISE at a single site is hence not only temporally but by using information about the spectral properties of the pending rock types without the need for other camera systems also spatially transferrable. This assumption will be further evaluated in future studies with more test sites.

We have now underlined the importance of a locally adapted NDSI threshold calibration for Landsat snow cover maps at the

two presented catchments. However, the detected NDSI threshold dependency automatically leads to the question if the need for threshold adaption is also necessary for coarser resolution satellite snow cover maps. This is of special interest as MODIS snow cover products are today the most frequently applied satellite snow cover maps. They are based on the NDSI technique and the 0.4 threshold and have a spatial resolution of 500 m. Hence, we aggregated the Landsat snow cover maps using calibrated and standard NDSI threshold values from 30 m to 90 m, 210 m, 510 m, and 990 m resolution. It can be seen that the

SCA deviation between standard and calibrated snow cover maps diminishes for coarser resolution data. Figure 10 a outlines this error reduction with spatial aggregation for a Landsat 7 scene of Vernagtferner catchment on 16 September 2011. Figure 10 b shows the simultaneously captured photograph used for calibration. Figure 10 c underlines this finding by depicting the spatial resolution at which standard and calibrated snow cover maps become identical for the 65 cases investigated in the two catchments. The aggregation step to 510m is thereby of major importance as more than 90% of SCA maps for our investigation

period and study become identical at this pixel size. Thus, using the standard threshold of 0.4 seems to be accurate in case of the MODIS snow cover product with a pixel size of 500m. For applications at this scale, the additional effort using camera calibrated data only provides slight improvements and might rarely justify the effort. However, our new method using camera-calibrated data allows to set in value the higher resolution satellite data of the Landsat series and of the new Sentinel 2.

## 5 Conclusions

The study has revealed that using the standard threshold of 0.4 is adequate for satellite products with a pixel size of 500 meters and more. For higher resolution snow cover mapping, significant improvements in the quality of the snow cover maps can be achieved if a threshold is used which is variable in space and time. The clustering-based segmentation technique of Otsu is



producing results which are only slightly different from those of the standard threshold of 0.4 and do not indicate a need for a further adaption. However when compared to local images, the resulting differences are becoming obvious and could only be reduced by the presented camera-based technique. The long-term analysis of calibrated $NDSI_{thr}$ at two comparable high elevation sites has shown that large deviations from the 0.4 standard threshold exist. The calibrated optimal threshold values

span a range from 0.15 to 0.74 over the complete time series and can reach a difference of 0.45 between both observation sites at a single date. It was also shown that these differences in $NDSI_{thr}$ lead to significantly different SCA when compared to the standard of 0.4.

The $NDSI_{thr}$ at both sites have similar seasonal dynamics while scattering around different site-specific average values (0.28 at RCZ, 0.57 at VF). The difference between the average threshold values at the two sites could be related to the different

reflection properties of the rock types in the investigation areas (limestone at RCZ and gneiss at VF). The overall correlation coefficient between $NDSI_{thr}$ of both sites is low (r=0.17) which prohibits a date by date transfer of calibrated values from one catchment to the other.

In view of the validity of the standard threshold of 0.4 at the local scale it was found that relative SCA error margins of up to 24.1% were found for the standard threshold method when using 30m Landsat products. This is critical for any snow cover

mapping application and especially for model evaluation studies. We hence conclude that the application of a fixed NDSI threshold can lead to large uncertainties in the resulting snow cover products at least at the local scale. Consequently, local studies strongly need to account for the $NDSI_{thr}$ variability in space and time in order to guarantee high accuracy snow cover products. But, in case studies are carried out with sensors having a pixel size of 500 meters and more the advantage of a location dependent $NDSI_{thr}$ vanishes.

It was shown that site-specific single-date adaptations of the $NDSI_{thr}$ also do not lead to resilient results. The uncertainty introduced by a single measurement is not quantifiable and can lead to results worse than that achieved by using the standard value of 0.4. A quantitative calibration or visual derivation of the $NDSI_{thr}$ for a single date and its application to other dates is therefore jeopardous.

The approximation of the $NDSI_{thr}$ over a simple seasonal model fitted to the calibrated $NDSI_{thr}$ at the respective site has

shown improvements instead. The achieved model was able to reduce the error in the SCA prediction by 50% when compared to the standard threshold method. Nevertheless, a fundamental data pool of in situ information covering the dynamic over the year as well as the range of possible $NDSI_{thr}$ within a season is needed for calculating this relation. Finally, it was shown that the fitted model parameters are also spatially transferable if an additional term accounts for the background radiation of the different rock types. This is possible without in situ measurements by utilising the constant NDSI reflectance differences of

the rock surface in the respective catchments. However, this needs to be further tested at more sites. Future studies will hence use the existent webcam infrastructure in the European Alps as well as camera systems installed worldwide at the INARCH network sites (Pomeroy et al., 2015) for the generation of numerous calibrated $NDSI_{thr}$. The observed threshold values will serve as operational source for applicable $NDSI_{thr}$ and will allow to evaluate the presented temporally and spatially variable prediction approach of $NDSI_{thr}$. In case of a successful evaluation, the presented scheme allows for an objective and



reproducible derivation of the $NDSI_{thr}$ value for any given satellite scene. This is a large advantage as the threshold is up to now often set intuitively or assumed as constant which does neither conform to the complexity of the models evaluated on basis of NDSI based snow cover maps nor to the needs of the models which are assimilating these maps.

**Acknowledgements**

5  This work was funded by the Austrian Science Fund (I 2142-N29), the doctoral scholarship program of the German Federal Environmental Foundation (DBU), the Helmholtz Research School Mechanisms and Interactions of Climate Change in Mountain Regions (MICMoR) and has additionally received a fundamental support of the Environmental Research Station Schneefernerhaus (UFS) in course of the Virtual Alpine Observatory (VAO). The Commission for Glaciology of the Bavarian Academy of Sciences and Humanities has kindly provided data of Vernagtferner. We want to thank the crew of the UFS

10 (Markus Neumann, Dr. Till Rehm and Hannes Hiergeist) for supporting this peace of research by hosting the authors and maintaining the camera system. Thomas Werz and Michael Weber have also supported the research by temporally maintaining the camera system. Relevant data can be made available by the authors.



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





**Table 1.** Basic statistic measures of the automatically derived NDSI threshold time series at RCZ and VF using the Otsu segmentation method and the camera-based calibration method.

| Site | Automatically derived NDSI threshold values | | | | | | | | | |
|------|--------|------|--------|------|--------|------|--------|------|--------|------|
| | Mean | | Standard Deviation | | Max | | Min | | Spread | |
| | camera | Otsu | camera | Otsu | camera | Otsu | camera | Otsu | camera | Otsu |
| RCZ | 0.28 | 0.36 | 0.07 | 0.04 | 0.39 | 0.45 | 0.15 | 0.29 | 0.24 | 0.16 |
| VF | 0.57 | 0.41 | 0.09 | 0.04 | 0.74 | 0.47 | 0.35 | 0.33 | 0.39 | 0.14 |



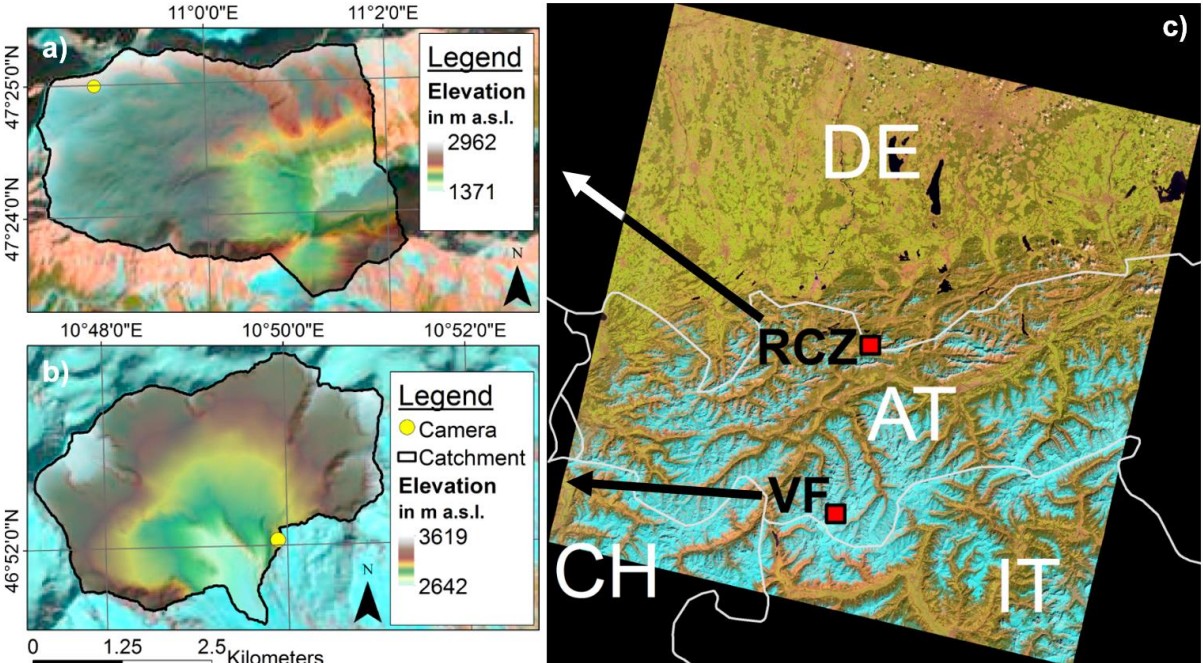

**Figure 1:** The figure shows the two test sites used in this study as well as their location within a Landsat scene. Both have indicated the camera location in yellow, the catchment area outlined in black and the digital elevation model (DEM) superimposed on a Landsat Look image. **a)** Research Catchment Zugspitzplatt (Germany), **b)** Vernagtferner catchment (Austria), **c)** Landsat scene (Landsat Look image, WRS2 path 193, row 27) which contains both sites.





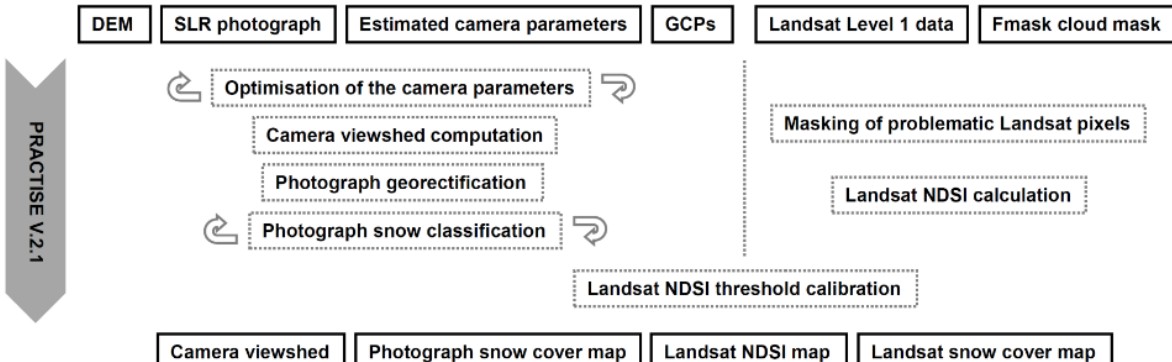

**Figure 2:** Input and output data as well as the workflow of PRACTISE (version 2.1) to generate the calibrated NDSI snow cover maps from Landsat data are depicted here (from Härer et al., 2016).




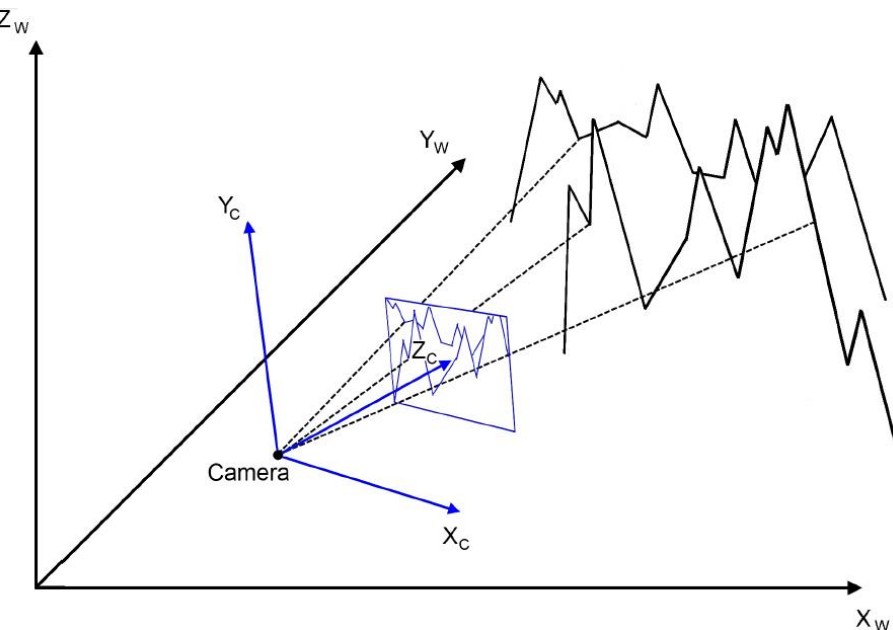

**Figure 3:** Schematic relationship between the camera location and orientation, and the two-dimensional photograph (blue) and three-dimensional real world (black).





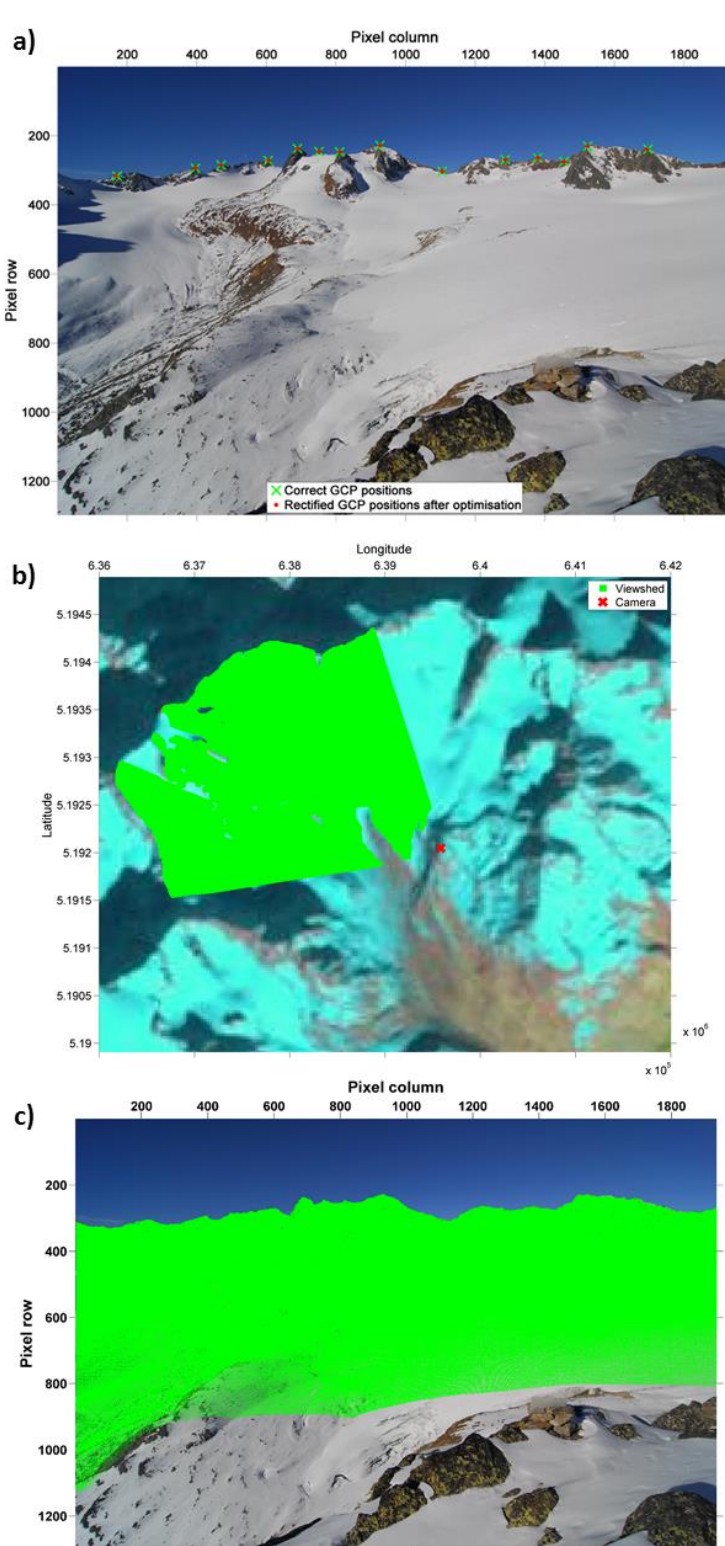



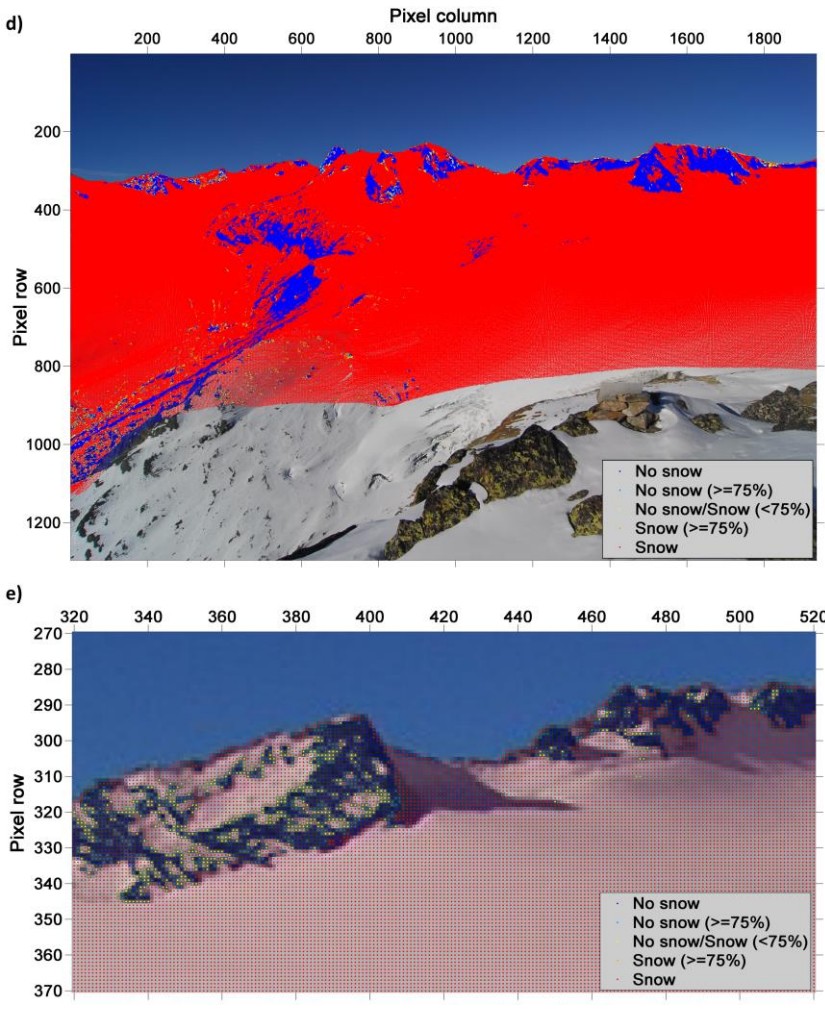

**Figure 4:** Internal processing steps within a single PRACTISE evaluation are shown for a photograph of VF on 17 November 2011. The figures chronologically show the routines for the photograph processing in PRACTISE which are **a)** the optimisation of the camera location and orientation using ground control points, **b)** the performed viewshed analysis from the resulting camera location and orientation, **c)** the projection and **d)** the classification of visible DEM pixels. More detail of the PCA based classification result in **d)** can be seen in an enlarged view in **e).**



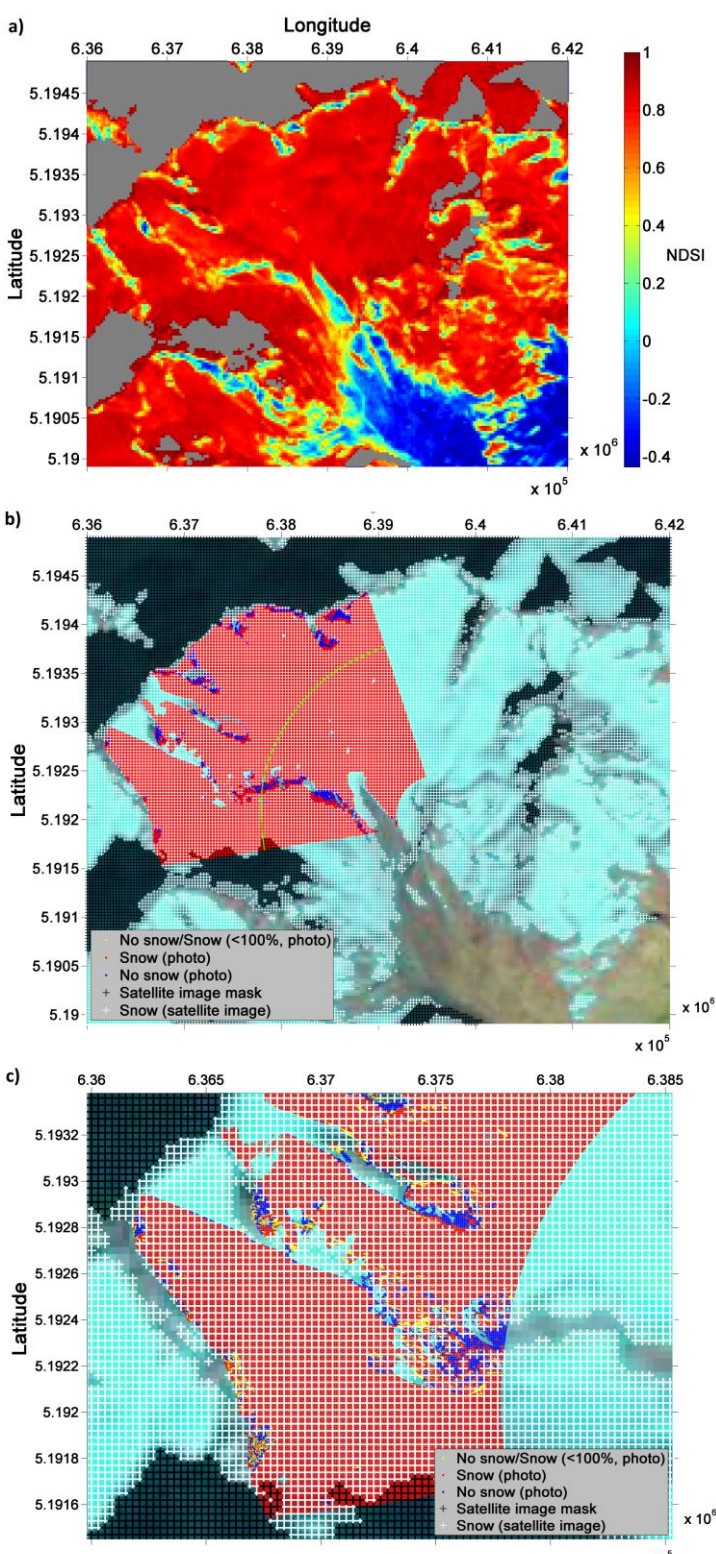

**Figure 5:** We outline here the internal processing steps within the remote sensing routines of PRACTISE. The Landsat NDSI map from 17 November 2011 is shown in **a)**. Clouds and shadows (grey areas) are excluded using fmask. The photograph and satellite snow cover map derived from the PRACTISE evaluation are superimposed on the Landsat Look image of 17 November 2011 in **b)**. Snow is depicted in red for the photograph snow map and white for the satellite snow map. The lower areas at VF (south-east of the green line in **b)**) were excluded from the complete analysis as the combination of strong glacier retreat at VF and temporal difference between some analysis dates and the DEM recording dates resulted in a discrepancy of real elevations and DEM in the lower catchment areas that affected $NDSI_{thr}$ calibration results. The cutout in **c)** clarifies which photographed areas are part of the analysis and additionally underlines the high agreement between photograph and satellite snow cover map.





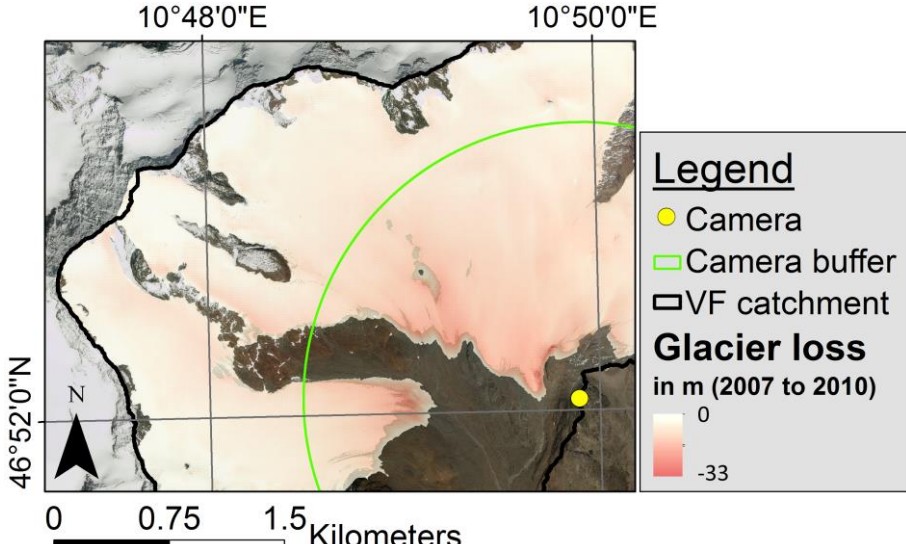

**Figure 6:** Glacier retreat from 2007 to 2010 causes a loss in elevation of up to -33m at VF. The green line depicts the buffer distance around the camera which was excluded from the analysis due to significant glacier loss which in turn lead to geometric inaccuracies in the photograph rectification and incorrect NDSI threshold calibration results.



**a)**

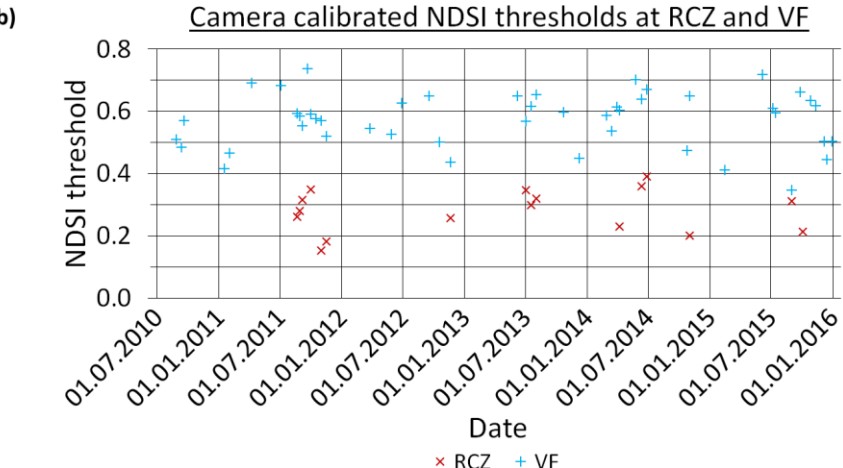

**Figure 7:** The Figure displays in **a)** the complete time series of adjusted NDSI thresholds using the Otsu segmentation method (circles, erroneous thresholds as squares) at RCZ (red) and VF (blue) and depicts in **b)** the camera calibrated NDSI thresholds at these two sites utilising ground-based photographs as in situ measurements (blue pluses for VF and red crosses for RCZ). Relative SCA changes at RCZ and VF resulting from the application of the standard instead of the camera calibrated reference NDSI threshold are shown in **c)**.

**b)**

**c)**

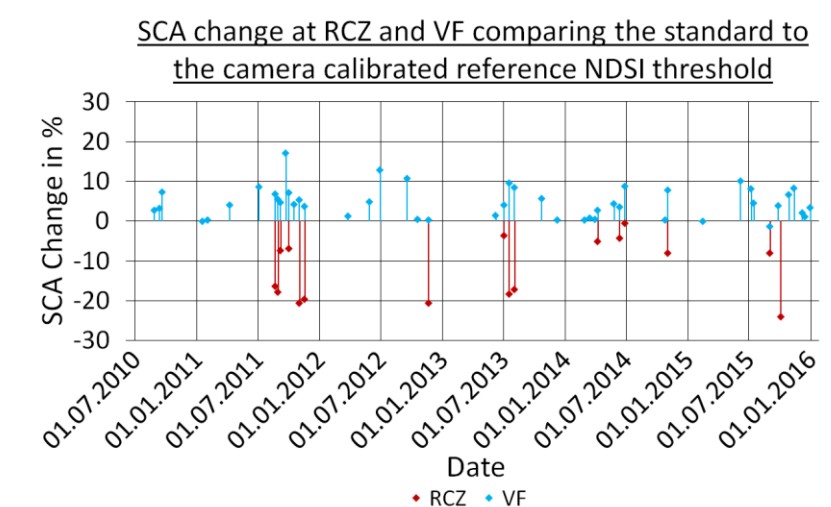



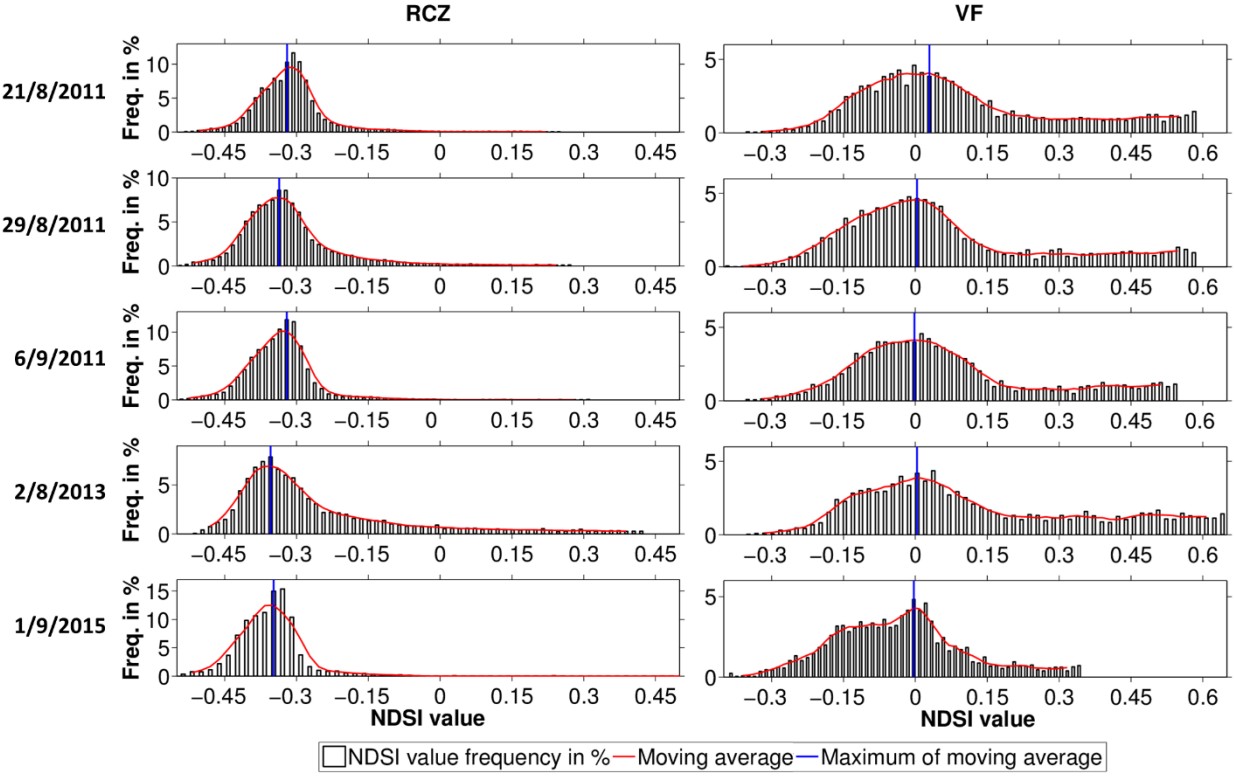

**Figure 8:** Representative NDSI reflectance values for the rock surfaces in RCZ and VF catchment are determined using frequency histograms of the snow-free bare rock NDSI values for five summer dates. These are then smoothed applying a moving average of 5 histogram classes. The maxima of the smoothed histograms are stable for each catchment and the investigated dates and result in mean NDSI values for rock surfaces at RCZ of -0.34 and at VF of 0.01.



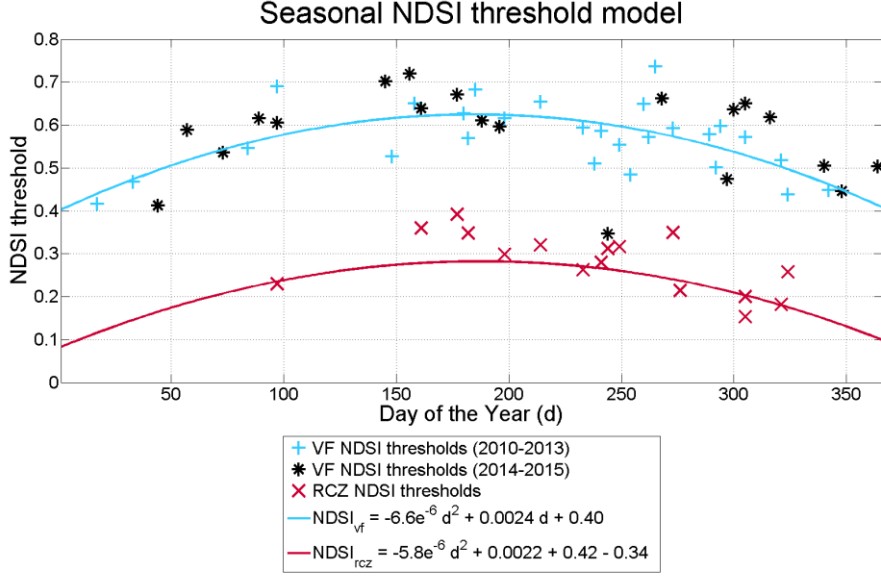

**Figure 9:** Estimates of NDSI threshold values at VF are predicted for each day of the year by a quadratic polynomial model ($NDSI_{vf}$, blue line) which was fitted to the calibrated NDSI thresholds between 2010 and 2013 ($NDSI_{thr}$, blue pluses). The coefficient of determination ($r^2$) of this model is 0.45 and the root mean square error (RMSE) is 0.06. The black stars represent the $NDSI_{thr}$ from 2014 to 2015 at VF used for evaluation of $NDSI_{vf}$. Additionally, a $NDSI_{thr}$ prediction model for RCZ ($NDSI_{rcz}$, red line) is defined by a quadratic polynomial model fitted to the complete time series of calibrated $NDSI_{thr}$ at VF (blue pluses and black stars, $r^2=0.36$, RMSE=0.07) and an additional term of -0.34 to account for the NDSI reflectance difference between the different rock surfaces at RCZ and VF. $NDSI_{rcz}$ is evaluated against the calibrated $NDSI_{thr}$ of RCZ (red crosses).



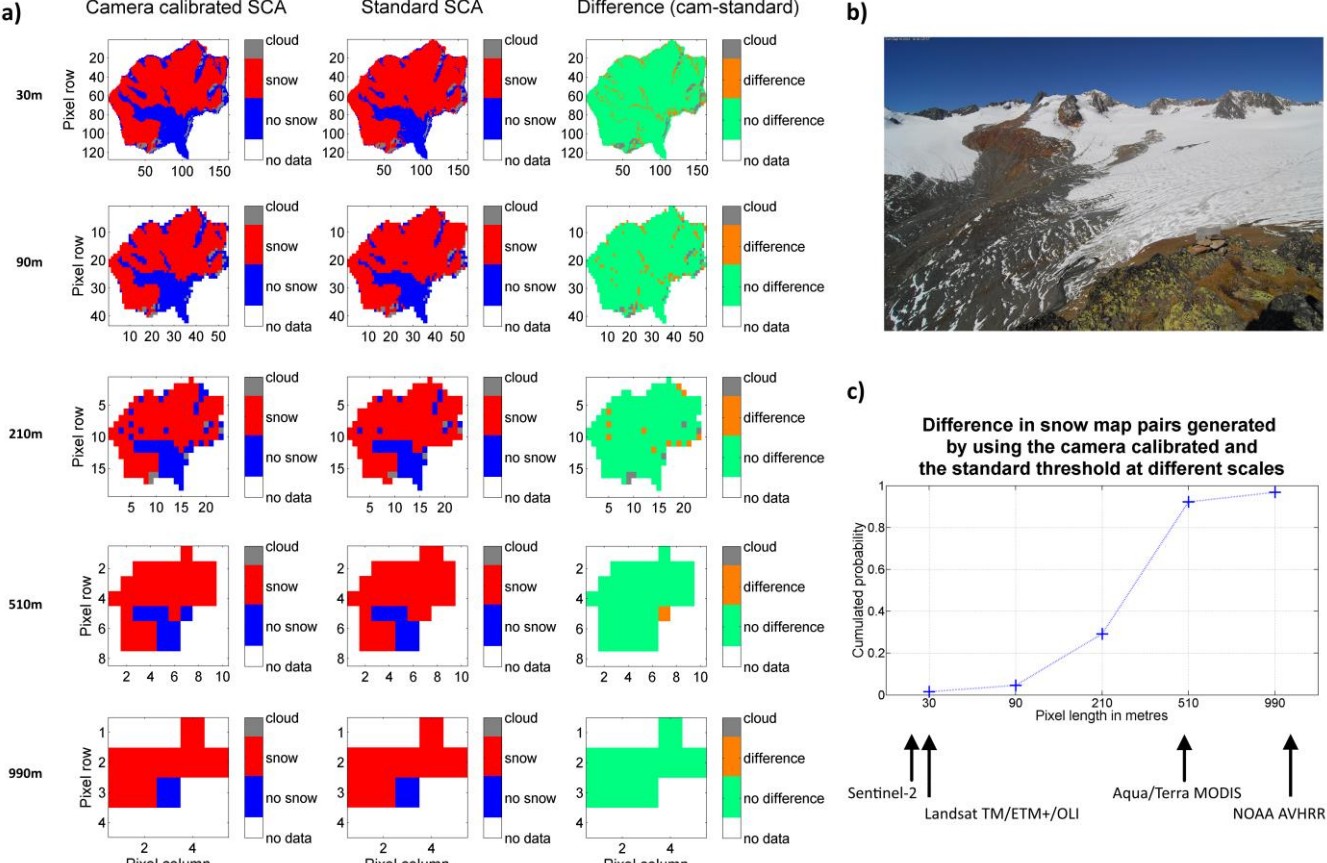

**Figure 10:** At VF, we exemplarily show in **a)** the effect of scaling to NDSI based snow cover products for a Landsat 7 scene at 16 September 2012. The first column outlines the camera calibrated SCA, in the second column the standard threshold SCA is depicted, and in the third column their differences at VF are presented. The different rows show different scaling factors, starting from the top with the original resolution and a factor of 1 (30 m) to 3 (90 m), 7 (210 m), 17 (510m) and at the bottom a factor of 33 (990 m). The concurrent photograph in **b)** depicts the snow situation at VF in our example. The analysis of all investigation dates in **c)** shows that camera calibrated and standard threshold snow cover maps become more and more identical with lower resolutions. The positive effect of the camera calibration for Landsat and presumably Sentinel-2 data thus diminishes for pixel sizes of 500 m and higher and hence for snow cover products derived from the MODIS or the AVHRR sensor.