# Peer review of "On the need of a time and location dependent estimation of the NDSI threshold value for reducing existing uncertainties in snow cover maps at different scales"

_The Cryosphere, 2017_

## Referee Comment (RC1) · Anonymous Referee #1 · 16 Oct 2017

On the need of a time and location dependent estimation of the NDSI threshold value for reducing existing uncertainties in snow cover maps at different scales

Stefan Härer, Matthias Bernhardt, Matthias Siebers, Karsten Schulz

The authors present an interesting study focusing on measurement of normalized-difference snow index (NDSI) using ground-based photogrammetric methods. They take a relatively newly developed methodology (PRACTISE v.2.1: previously published by the authors in GMD) and apply it to two high elevation catchments in the Euro-

pean Alps to estimate very high resolution (1m) NDSI. These high-resolution outputs of PRACTISE are then compared to lower resolutions (30 to 990 m) to compare with Landsat and MODIS imagery, amongst other potential satellite products. This is a very worthwhile study as the eventual application of this technique across multiple high mountain basins could provide essential information about the rate of decrease in mountain snowpacks in a warming world.

While the main thrust of the scientific messages are understandable and important, there remains some more work that needs to be done to increase the clarity of the argument. I make the following suggestions for consideration that I hope would benefit the paper.

1. The current manuscript introduces the application of PRACTISE, but does not provide a detailed enough description to be able to fully understand how it works. Considering there are already two detailed papers on PRACTISE by the authors in GMD, I would hope citations to them could provide the reader with a satisfactory description. Then this manuscript could be refined to provide more focus on the results of the RCZ / VF comparison, and greater detail on the scaling question. Figures 2 through 5 could be removed to focus more on the results of this study (Fig 6 through 9) and expand analysis around figure 10 – which is of great interest, but under-analyzed in the current manuscript.

2. The comparison between RCZ / VF is robust and well quantified (Fig 7). The influence of rock reflectance is well described (Fig 8) and provides a valuable process basis to the quadratic relationships (Fig 9). However, the scaling question, while well illustrated by the example of 16 September (Fig 10a), suggests that all data have been used to create relationships presented in (Fig 10c). How was Fig 10c constructed and what does the 'cumulated probability' mean in this context? Can you show that the increase in the identical nature of snow cover maps is not simply a function of decreasing number of snow maps pairs (again this links to greater clarification as to what is meant by cumulated probability). If this relationship is statistically robust, could more

be made of this message, as understanding the influence of measurement resolution by satellite imagery is important to understanding the fate of snow and ice in small glacierized basins.

3. Currently there are deficiencies in the terminology and language used throughout the manuscript. These require detailed attention. A non-exhaustive list of examples can be found below:

Pg1, ln19 – Quantify how different the statistically insignificant correlation was to the standard threshold.

Pg1, ln20 – what is the 'another literature value'? State it here.

Pg1, ln 21 – replace 'case' with 'cases where'.

Pg2, ln5 – 'precipitation water' – just say 'precipitation'?

Pg2, ln 10-13 – avoid single sentence paragraphs. Change this throughout the manuscript.

Pg2, ln28 – 'In this context' is superfluous and could be removed.

Pg3, Ln 23 – 'built up' is poor terminology for geological composition

Pg3, ln 25 – 'pending' is strange terminology. Do you mean 'underlying rock' or 'substrate'

Pg3, ln 29 – no need for 'for' in the statement 'guarantees for comparable'

Pg4, ln 1 – do you mean 'dates' rather than 'cases'? Stick to constant terminology.

Pg4, ln 4 – is 'rectifaciton' as spelling error?

Pg5, ln 1 – no need for 'It has to be mentioned that'

Pg5, ln 22 – remove 'an'

Pg5, ln 30 & 32 – remove 'used'

Pg5, ln 32 – 'misclassificied' is misspelt

P6, ln 4-6 – tenses are used interchangeably. Suggest sticking to past tense consistently throughout the methods section

Pg6, ln 9 – remove 'thereby'

P6, ln 13 - no need for 'It has to be mentioned that'

Pg7, ln 10 – what does 'underline' mean in this context, I think the wrong word is being used here.

Pg7, ln 20 – if statistically significant, then present the stats here (r-value and p-value).

Pg8. Ln 17 – remove 'in percents'

Pg8, ln24 – what chose a threshold of 0.7? Provide some justification.

Pg10, ln 11 – what is a 'date by date transfer'?

Pg10, ln 23 – 'jeopardous' is probably not the correct term to use here. 'inappropriate' or something similar may be better.

Figures: Use of titles within figures and sub-figures is unnecessary (e.g. above each sub-figure in Fig 7). Instead use the associated caption to clearly describe each figure. Often current captions are a mix of methods and results, rather than sticking to the bare minimum need to adequately describe what is presented. The main body of the text should instead be used to explain methodological procedures and results.

---

## Referee Comment (RC2) · S. Gascoin (Referee) · 2 Nov 2017

The NDSI is a spectral index that is commonly used to map the snow cover extent from optical multispectral satellite imagery. Here, the authors evaluated the stability of the NDSI threshold to map the snow cover in two alpine sites from a time series of Landsat images. The paper reads rather well. I found that the wording is sometimes a bit awkward but I'm not an expert in English grammar!

The study is interesting, but I suggest to enhance the analysis and discussion of the

results to demonstrate the added-value of this paper with respect to a previous paper by the same authors (Härer et al. 2016 GMD). I made some suggestions below. I hope the authors will find my comments useful.

1) The authors choose not to apply atmospheric correction to the Landsat scenes. Absorption and scattering by the air molecules and aerosols can have an impact on the NDSI, although atmospheric absorbance is typically low in the SWIR and Green wavelengths, which are used to compute the NDSI (especially in high elevation areas). However, this shall be better discussed since the authors justify their study by the fact that the 0.4 NDSI threshold is used for the MODIS snow products ("This is of special interest as MODIS snow cover products are today the most frequently applied satellite snow cover maps"). If the authors referred to NASA's MOD10/MYD10 snow products, the atmospheric correction is applied before computing the NDSI. In addition, the NDSI threshold is not applied anymore in the latest collection 6. More importantly, the lack of atmospheric correction cast doubts on the significance and the transferability to other sites of the "newly developed calibrated quadratic polynomial model which is accounting for seasonal threshold dynamics". The authors should clarify this to avoid confusing readers who are not familiar with satellite imagery processing.

2) The authors give too much details about the PRACTISE software, which was used to rectify the photographs from the time lapse cameras, whereas it was already described in another journal (for instance Fig. 2 was already published in Härer et al. 2016; Fig. 3 and Fig. 4 further illustrate the PRACTISE workflow and are not useful in my opinion). An important step for this study is rather how these camera snow maps were resampled to the Landsat resolution and it is missing. Indeed, camera snow maps have a sub-meter ground sampling distance. As a result, it is likely that some Landsat pixels were classified as "snow" from the camera images, while they were actually not 100% snow covered in the camera images.

3) The literature review in the introduction was a bit overlooked. The authors state that "The used snow-cover mapping approaches can be grouped into three categories:

manual interpretation, classification-based, and index-based methods" but there are other approaches based on spectral unmixing. The proposed "geology dependent offset" is the result of a well-known phenomena (e.g. Kaufman et al. 2002, GRL), and is similar to the NDSI_0 method developed by Chaponnière et al. (2005, IJRS); it can be also seen as an extreme simplification of a spectral mixture analysis used in other MODIS snow products (e.g., Sirguey et al., 2009 RSE; Painter et al., 2009 RSE).

4) Overlap with Härer et al. (2016 GMD). In a previous paper, the authors already showed the results of the NDSI threshold calibration on three Landsat scenes using the same method. Here, the authors extend this approach to a time series of Landsat images, which is a good idea I think. The authors obtain a (weak) seasonal cycle in the calibrated NDSI threshold value. Given that an important insight of this TCD paper was already introduced by Härer et al. 2016 ("A spatial and temporal adjustment of NDSI thresholds is therefore important to ensure optimum results in the snow cover mapping"), I do not think that the "investigation of the reasons of this effect is beyond this study". The authors could test if the calibrated NDSI_tr is correlated to the solar zenith and azimuth angles. In addition, the authors did not consider the hypothesis that this seasonal cycle may be due to inaccurate snow detection in the camera images. What is the bit depth of the camera images? Snow detection from terrestrial camera imagery is difficult in shaded slopes especially from 8-bits RGB pictures. The reported accuracy (below 5% misclassified pixels from visual inspection) can lead to significant changes in the NDSI, which are probably within the range of the calibrated NDSI threshold variability? This could be tested by excluding shaded areas before computing the accuracy of the Landsat snow masks. Another source of error that was not discussed is the one due to the geometric distortion between oblique images and nadir-looking satellite images.

Specific comments

P1L12: Earth not earth

P3L22: glacierized not glaciered

P3L23-25: It could be useful to show the spectral profiles of the snow-free substratum (limestone is more reflective in the visible range than gneiss).

P4L13: I do not think that this statement is true "no atmospheric correction is applied (..) the majority of studies that apply the NDSI for snow cover mapping". Many studies use the MOD10 snow products, or TMSCAG for Landsat, which use surface reflectances.

P3L17: NDSI and NDSI_thr are written in equation mode, sometimes in plain text.

P4L9: photographs not photographies.

What is the acquisition time of the camera? is it synchronous to Landsat overpass time?

P4L31: did you find a difference in the results between Landsat 8 and the Landsat 5/7? Landsat 8 instrument has higher radiometric resolution which improves snow classification in mountains (less saturation, higher SNR in shaded slopes).

P6L8: Note that this metric is usually referred to as accuracy and may not be a robust performance measure when the number of a class is much greater than the number of the other one.

P7L29: if vertical, these rock faces are not visible in images captured by nadir-looking sensors like Landsat 7.

P7L31: "NDSI reflectances" does not make sense

P8L10-12: this sentence is not clear to me.

P8L31: fitted against what? Day of year I think.

P9L18: why not using real MODIS images instead?

P9L25: if I understand well, the resampling to 500m has increased the optimal threshold value. Can you think of an explanation?

Fig. 5: rainbow colormaps are not recommanded (Borland 2007). I am also surprised by the choice of the projection (plate carrée? non-equidistant projections are not recommended for this kind of maps)

---

## Author Response (AR1)

**Answer to the review of the anonymous referee #1:**

**General comments:**

**1.** The current manuscript introduces the application of PRACTISE, but does not provide a detailed enough description to be able to fully understand how it works. Considering there are already two detailed papers on PRACTISE by the authors in GMD, I would hope citations to them could provide the reader with a satisfactory description. Then this manuscript could be refined to provide more focus on the results of the RCZ / VF comparison, and greater detail on the scaling question. Figures 2 through 5 could be removed to focus more on the results of this study (Fig 6 through 9) and expand analysis around figure 10 – which is of great interest, but under-analyzed in the current manuscript.

**Answer to 1.** We thank you for the comment and we will revise the paragraphs on the application of PRACTISE in this study to clarify the workflow used in this study by including more references to our previous papers on PRACTISE. However, we also think that the figures 2 to 5 are important here as they outline the workflow in PRACTISE graphically. This makes it simpler for non-experts to understand the processing steps taken in this specific study and it was a recommendation of the editor to our initial submission. Moreover, with these figures, readers are not obliged to read our previous papers where they probably face much more options and detail than they need. The second part of your comment is proposing to expand the analysis on the scaling effects between different NDSI thresholds. We agree with you on this point (see the answer to comment 2).

**Manuscript changes to 1.**

p.6, l1 to p.7, l5: "In a first step information about the camera location and orientation was needed for georectification of the photography. This information was automatically optimized by using ground control points (GCPs, Fig. 4a; Sect. 3.3 in Härer et al., 2013). The calculated viewpoint and viewing direction were by default used to perform a viewshed analysis (Fig. 4b; Sect. 3.1 in Härer et al., 2013). The viewshed was needed for an identification of areas which were visible from the viewpoint and which were not obscured by topographical features or within a user-specified buffer area around the camera. The respective DEM pixels were then projected to the photo plane (Fig. 4c; Sect. 3.2 in Härer et al., 2013).

Now, the snow classification module was activated to distinguish between snow covered and snow-free DEM pixels (Fig. 4d). Two major procedures were available for classification: a statistical analysis which was using the blue RGB band (Salvatori et al. 2011; Sect. 3.4 in Härer et al., 2013) and a principal component analysis (PCA) based approach (Sect. 3.1 in Härer et al. 2016). The first was used for shadow-free scenes, the second for scenes with shaded areas. Section 3.4 in Härer et al. (2013) gives more insights into a third manual option if none of the two classification routines could be applied successfully. The photograph snow cover maps did have even in the case that an insufficient classification algorithm was used for a specific situation less than 5% misclassified pixels in the worst case region of the photograph in Chapt. 4 in Härer et al. (2013). It was also shown in an earlier publication that the classification of shadow-affected photographs are of the same quality as photographs without shadows (Chapt. 4 in Härer et al., 2016). As for this study, every classified image

was visually inspected and no major snow classification errors comparable to our worst case example in the previous publication were found, we expect a relative misclassification error of 1%. For this example photograph, the snow classification algorithm utilizing a principal component analysis (PCA) was selected to account for the shadow-affected areas in the upper left part of the photograph (Fig. 4d, enlarged view in Fig. 4e).

After the photograph rectification and classification, the remote sensing routine of PRACTISE began with the identification of satellite pixels that spatially overlap with the photograph snow cover map (Sect. 3.2 in Härer et al., 2016). The used photograph and satellite image were thereby recorded within one (RCZ) to three (VF) hours. Moreover, a cloud- and shadow-free satellite image is generated by using fmask (Zhu et al., 2015). The needed NDSI map was calculated in accordance to Eq. (1) by PRACTISE (Fig. 5a).

If both, the NDSI satellite map and the corresponding high resolution photograph snow cover map were processed, an iterative calibration of the NDSI threshold value was started. The Landsat image was thereby resampled to the finer resolution of the photograph in the calibration to avoid losing any information by the aggregation of the photograph snow cover map. The best agreement between the local scale (photograph) and the large scale (Landsat) snow cover map was detected by maximizing the accuracy which is the ratio of identically classified pixels to the overall number of photograph-satellite image pixel pairs $n$ (Aronica et al., 2002):

$$F = \frac{(a+d)}{n},$$
(2)

$a$ represents the number of correctly identified snow pixels and $d$ the same for no snow pixels. $F$ is between 0 and 1 and becomes 1 for a perfect agreement between the two images.

The calibrated NDSI threshold was finally applied to the Landsat data with 30m pixel size to generate the final Landsat snow cover map. Figure 5b shows the resulting satellite snow cover map superimposed on the photograph snow cover map and a Landsat Look image. A cutout is shown for more detail in Fig. 5c. ”

**2.** The comparison between RCZ / VF is robust and well quantified (Fig 7). The influence of rock reflectance is well described (Fig 8) and provides a valuable process basis to the quadratic relationships (Fig 9). However, the scaling question, while well illustrated by the example of 16 September (Fig 10a), suggests that all data have been used to create relationships presented in (Fig 10c). How was Fig 10c constructed and what does the 'cumulated probability' mean in this context? Can you show that the increase in the identical nature of snow cover maps is not simply a function of decreasing number of snow maps pairs (again this links to greater clarification as to what is meant by cumulated probability). If this relationship is statistically robust, could more be made of this message, as understanding the influence of measurement resolution by satellite imagery is important to understanding the fate of snow and ice in small glacierized basins.

**Answer to 2.** We really appreciate that you value our work presented using the figures 7 to 10. With respect to figure 10c, you are firstly right that all analysed data at Vernagtferner and Zugspitze is used to generate the graph. The term 'cumulated probability' in figure 10c might be unclear, we will change it to the 'number of identical snow cover maps' where the total number is 63. The figure hence answers

the question how many of the snow maps generated with the calibrated and the standard threshold are identical in the complete catchment areas for the different pixel sizes between 30 m and 990 m. For example, 58 of the 63 snow maps (over 90%) are completely identical at a pixel size of 510 m.

Now, we come to the final part of your question where you asked if the identical nature of snow cover maps is not simply a function of decreasing number of snow map pairs and if this relationship is statistically robust. We can neither agree nor deny this statement from our data. We thought about this effect and our data shows often increase but also at other dates decrease in agreement for coarser resolutions. The simple reason is that one pixel being different has a stronger relative effect with coarser resolution as our catchments are small and the number of pixels becomes low. We therefore decided to not look at the relative increase in agreement of the calibrated and standard snow cover maps with larger pixel sizes but to focus on the pixel size at which the snow cover maps become completely identical as this is independent of relative changes with aggregation. Fig. 10 c outlines when total agreement of the snow cover maps for the different NDSI thresholds was found.

**Manuscript changes to 2.**

Figure 10

[Figure]

p.11, l1 to 16: "Our aggregation experiment of the Landsat snow cover maps for the different $NDSI$ thresholds shows that the SCA deviation between standard and calibrated snow cover maps diminishes for coarser resolution data in most cases. Figure 10 a outlines this error reduction with spatial aggregation for a Landsat 7 scene of Vernagtferner catchment on 16 September 2011. Figure 10 b shows the simultaneously captured photograph used for calibration. We however cannot draw an absolute conclusion from fig. 10 a that the difference in snow cover maps between the different thresholds is always reduced with a coarser resolution. The simple reason is that with larger pixel sizes,

the number of pixels in the catchment becomes lower and the relative weight of a pixel being different for different thresholds has a larger relative weight. Therefore, we decided to investigate at which spatial resolution the standard and calibrated snow cover maps become identical for the 63 cases investigated in the two catchments. This variable is absolute and thus independent of relative weights and changes with spatial aggregation. The aggregation step to 510m is thereby of major importance as more than 90% (58 of 63) of SCA maps become identical at this pixel size. Thus, using the standard threshold of 0.4 instead of the higher $NDSI$ thresholds at VF and the lower $NDSI$ values at RCZ seems to be accurate in most cases with a pixel size of 500m. For applications at this scale, the positive effect of using camera calibrated data diminishes and might rarely justify the effort."

**Specific comments:**

Note: We do not show each of the manuscript changes for the specific comments here as the changes are obvious by the answer. Nonetheless, the changes will be denoted in the new manuscript.

**Pg1, ln19**: Quantify how different the statistically insignificant correlation was to the standard threshold.

**Answer:** We add the correlation coefficient here.

**Pg1, ln 20**: what is the 'another literature value'? State it here.

**Answer:** The other literature value is the locally optimized 0.7 threshold value of Maher et al. (2012) which was also found for single events at Vernagtferner. We add this to the abstract.

**Pg1, ln 21:** replace 'case' with 'cases where'.

**Answer:** Thank you for the correction.

**Pg2, ln 5:** 'precipitation water' – just say 'precipitation'?

**Answer:**, We delete the word 'water'.

**Pg2, ln 10-13:** avoid single sentence paragraphs. Change this throughout the manuscript.

**Answer:** You are right, we change the single sentence paragraphs in our manuscript.

**Pg2, ln28:** 'In this context' is superfluous and could be removed.

**Answer:** We remove it, thank you.

**Pg3, Ln 23:** 'built up' is poor terminology for geological composition

**Answer:** We agree and change the terminology.

**Pg3, ln 25:** 'pending' is strange terminology. Do you mean 'underlying rock' or 'substrate'

**Answer:** Thank you, we use 'underlying rock' now.

**Pg3, ln 29:** no need for 'for' in the statement 'guarantees for comparable'

**Answer:** Thank you for the correction.

**Pg4, ln 1:** do you mean 'dates' rather than 'cases'? Stick to constant terminology.

**Answer:** We totally agree, we change it.

**Pg4, ln 4:** is 'rectifaciton' as spelling error?

**Answer:** Yes, you are right, it should be 'rectification'.

**Pg5, ln 1:** no need for 'It has to be mentioned that'

**Answer:** We agree and remove it accordingly.

**Pg5, ln 22:** – remove 'an'

**Answer:** Thank you for the correction.

**Pg5, ln 30 & 32:** remove 'used'

**Answer:** We remove 'used' in both sentences.

**Pg5, ln 32:** 'misclassificied' is misspelt

**Answer:** Thank you for finding this spelling error.

**P6, ln 4-6:** tenses are used interchangeably. Suggest sticking to past tense consistently throughout the methods section

**Answer:** You are right, we change the tense in the methods section to past tense where appropriate.

**Pg6, ln 9:** remove 'thereby'

**Answer:** We correct it.

**P6, ln 13:** no need for 'It has to be mentioned that'

**Answer:** You are right, we remove it.

**Pg7, ln 10:** what does 'underline' mean in this context, I think the wrong word is being used here.

**Answer:** This is true, we wanted to clarify that the minimal differences between the Otsu method and the standard threshold does not justify the additional effort needed for the Otsu method. We use 'justify' now."

**Pg7, ln 20:** – if statistically significant, then present the stats here (r-value and p-value).

**Answer:** We change 'significantly weaker' to 'weaker' as the expression was not meant in a statistically quantitative way here. It is a qualitative statement.

**Pg8. Ln 17:** – remove 'in percents'

**Answer:** Thank you for the correction.

**Pg8, ln24:** – what chose a threshold of 0.7? Provide some justification.

**Answer:** We have chosen the 0.7 threshold as it is in the range of plausible NDSI threshold values (0.35 to 0.7) that might result from a single date calibration at VF. Moreover, Maher et al. stated this value in their study area when they also calibrated the NDSI threshold only for one date due to the lack of additional data. We simply want to show here that a NDSI threshold calibrated at a single date does not give too much insight how the NDSI threshold might look like at other dates but at the same time gives investigators a false sense of confidence. We clarify it.

**Pg10, ln 11:** – what is a 'date by date transfer'?

**Answer:** The term was misleading. We wanted to state that we tested if the NDSI threshold calibrated at one catchment can be used at another catchment in the same Landsat scene. We rephrase the sentence.

**Pg10, ln 23:** – 'jeopardous' is probably not the correct term to use here. 'inappropriate' or something similar may be better.

**Answer:** Thank for the suggestion, we change it.

**Figures:** Use of titles within figures and sub-figures is unnecessary (e.g. above each sub-figure in Fig 7). Instead use the associated caption to clearly describe each figure. Often current captions are a mix of methods and results, rather than sticking to the bare minimum need to adequately describe what is presented. The main body of the text should instead be used to explain methodological procedures and results.

**Answer:** Thank you, we will change the figure captions and the respective text in the main body of the text.

**Answer to the review of S. Cascoin**

**General comments:**

**1)** The authors choose not to apply atmospheric correction to the Landsat scenes. Absorption and scattering by the air molecules and aerosols can have an impact on the NDSI, although atmospheric absorbance is typically low in the SWIR and Green wavelengths, which are used to compute the NDSI (especially in high elevation areas). However, this shall be better discussed since the authors justify their study by the fact that the 0.4 NDSI threshold is used for the MODIS snow products ("This is of special interest as MODIS snow cover products are today the most frequently applied satellite snow cover maps"). If the authors referred to NASA's MOD10/MYD10 snow products, the atmospheric correction is applied before computing the NDSI. In addition, the NDSI threshold is not applied anymore in the latest collection 6. More importantly, the lack of atmospheric correction cast doubts on the significance and the transferability to other sites of the "newly developed calibrated quadratic polynomial model which is accounting for seasonal threshold dynamics". The authors should clarify this to avoid confusing readers who are not familiar with satellite imagery processing.

**Answer to 1)** Thank you for this useful comment. You are right that we did not state the atmospheric correction of the MODIS snow cover product (MOD10/MYD10). And it is also true that the recently updated MODIS snow cover product collection does not use the fixed threshold of 0.4 anymore but uses a flag system in combination with a NDSI value of 0 as threshold. However, an own NDSI threshold can be used. Hence, the new algorithm gives more freedom to the users. This will probably lead to the situation that many users that cannot assess the value of the flag system or the best NDSI threshold for their scene might simply use the standard 0.4 threshold value again. In any case, we will add this information in the introduction to clarify the new situation for MODIS snow products to users unexperienced in this field of research as well as that we agree to add the statement to the manuscript that the developed quadratic approach might be only transferable to other high elevation areas at the moment and that further tests and probably an atmospheric correction of the data are needed if an application in lowland areas is planned.

**Manuscript changes to 1)**

p.2, l.29 to 34: "Accuracies in this range even though for the atmospherically corrected MODIS snow cover product (MOD10/MYD10) make $NDSI$ based snow cover products well accepted for global scale applications, but uncertainties have to be expected at the local scale (Härer et al. 2016). Moreover, the snow detection algorithm for the MODIS snow cover product changed in the latest collection 6. The algorithm now uses a NDSI threshold of zero together with a flag system to detect snow cover and users are encouraged to use their own NDSI threshold in the MODIS Snow Products Collection 6 User Guide if a binary snow cover map is wanted."

p.10, l.29 to 32: "However, the detected $NDSI$ threshold dependency automatically leads to the question if the need for threshold adaption is also necessary for coarser resolution satellite snow cover maps, for example, for a spatial resolution of 500 m or 1 km."

p.10, l.23 to 27: "This assumption and the transferability of the model is probably only true for high elevation areas. Even though that the $NDSI$ is an index which reduces the dependence on atmospheric conditions, an atmospheric correction might be necessary as well as more dynamic approaches that reflect the vegetation growth and senescence over the year for lowland areas. Hence, the approach needs to be further evaluated and developed in future studies with more test sites."

**2)** The authors give too much details about the PRACTISE software, which was used to rectify the photographs from the time lapse cameras, whereas it was already described in another journal (for instance Fig. 2 was already published in Härer et al. 2016; Fig. 3 and Fig. 4 further illustrate the PRACTISE workflow and are not useful in my opinion). An important step for this study is rather how these camera snow maps were resampled to the Landsat resolution and it is missing. Indeed, camera snow maps have a submeter ground sampling distance. As a result, it is likely that some Landsat pixels were classified as "snow" from the camera images, while they were actually not 100% snow covered in the camera images.

**Answer to 2)** We were encouraged to extend the details about the PRACTISE software and its application in its study by the editor before the discussion was opened. And we thank the editor now for this recommendation as we see the benefit that users interested in the approach but not in each detail of the algorithm of PRACTISE can easily follow the processing steps needed to calibrate the NDSI threshold of a photograph with this description. You are however right that we should add the resampling strategy used for the different spatial resolutions of the georectified photographs (1 and 5m) and of the Landsat satellite image (30 m). Moreover, we added more citations to our earlier publications on PRACTISE to clarify the workflow (see comment1 of the anonymous reviewer). To avoid losing any information, we used the finer resolution for calibration by resampling the Landsat resolution to the photograph resolution. The calibrated NDSI threshold is then finally applied to the Landsat pixels at their original resolution of 30m to generate the Landsat snow cover map which indeed will have mixed pixels.

**Manuscript changes to 2)**

p.6, l.27 to p.7, l4: "The Landsat image was thereby resampled to the finer resolution of the photograph in the calibration to avoid losing any information by the aggregation of the photograph snow cover map. The best agreement between the local scale (photograph) and the large scale (Landsat) snow cover map was detected by maximizing the accuracy which is the ratio of identically classified pixels to the overall number of photograph-satellite image pixel pairs *n* (Aronica et al., 2002):

$$F = \frac{(a+d)}{n},\qquad\qquad\qquad\qquad\qquad (2)$$

*a* represents the number of correctly identified snow pixels and *d* the same for no snow pixels. *F* is between 0 and 1 and becomes 1 for a perfect agreement between the two images.

The calibrated $NDSI$ threshold was finally applied to the original Landsat data with 30m pixel size to generate the final Landsat snow cover map."

**3)** The literature review in the introduction was a bit overlooked. The authors state that "The used snow-cover mapping approaches can be grouped into three categories: manual interpretation, classification-based, and index-based methods" but there are other approaches based on spectral unmixing. The proposed "geology dependent offset" is the result of a well-known phenomena (e.g. Kaufman et al. 2002, GRL), and is similar to the NDSI_0 method developed by Chaponnière et al. (2005, IJRS); it can be also seen as an extreme simplification of a spectral mixture analysis used in other MODIS snow products (e.g., Sirguey et al., 2009 RSE; Painter et al., 2009 RSE).

**Answer to 3)** We agree that our literature review in the introduction benefits from the references that you proposed. We thus include them in the revised manuscript.

**Manuscript changes to 3)**

p.2, l.13 to 19: "The used snow-cover mapping approaches can be grouped into four categories: manual interpretation, classification-based and index-based methods, and spectral mixture analysis. Manual interpretation as well as classification-based approaches are often used in local snow cover mapping studies. Both are out of the scope of this study as a need for expert knowledge and a high time-demand limit their applicability for large time series data. Spectral Mixture Analysis are also not in the focus of this study as they need an extensive spectral database for the different land surface components (Sirguey et al., 2009; Painter et al., 2009). These databases are usually not commonly available and only the final snow cover product can be downloaded (TMSCAG for Landsat and MODSCAG for MODIS)."

p.3, l.23 to 25: "Moreover, we present a seasonal model calibrated with the $NDSI$ threshold time series. The quadratic polynomial model can also be locally adapted by including a geology dependent offset which is comparable to earlier findings of Chaponnière et al. (2005)."

**4)** Overlap with Härer et al. (2016 GMD). In a previous paper, the authors already showed the results of the NDSI threshold calibration on three Landsat scenes using the same method. Here, the authors extend this approach to a time series of Landsat images, which is a good idea I think. The authors obtain a (weak) seasonal cycle in the calibrated NDSI threshold value. Given that an important insight of this TCD paper was already introduced by Härer et al. 2016 ("A spatial and temporal adjustment of NDSI thresholds is therefore important to ensure optimum results in the snow cover mapping"), I do not think that the "investigation of the reasons of this effect is beyond this study". The authors could test if the calibrated NDSI_tr is correlated to the solar zenith and azimuth angles. In addition, the authors did not consider the hypothesis that this seasonal cycle may be due to inaccurate snow detection in the camera images. What is the bit depth of the camera images? Snow detection from terrestrial camera imagery is difficult in shaded slopes especially from 8-bits RGB pictures. The reported accuracy (below 5% misclassified pixels from visual inspection) can lead to significant changes in the NDSI, which are probably within the range of the calibrated NDSI threshold variability? This could

be tested by excluding shaded areas before computing the accuracy of the Landsat snow masks. Another source of error that was not discussed is the one due to the geometric distortion between oblique images and nadir-looking satellite images.

**Answer to 4)** Thank you for this comment. We want to mention here that we used two cameras but only at a single catchment and 3 dates in our case study publication from 2016. We therefore were not sure what the spatial representativity of the calibrated NDSI threshold is within the same Landsat scene. Another question additional to the time series analysis that you mention was if we could use a single location for calibration and then use it for this Landsat scene? The study is therefore also completely new as a second catchment within the same Landsat scene is used for testing the spatial representativity within the scene. The systematic offset that was found, analysed and interpreted is thus also a major finding of the publication. Your thoughts on the reasons for variability are the same that we have. We also want to know what is driving the variability. However, this opens a really huge field of options that could be tested (e.g. albedo, snow grain size, snow age, …) which might need an extended experimental setup and testing all of these options would fill a complete publication on its own. So, it will be a task of our future work. We nevertheless agree that a correlation test for solar zenith and azimuth angles might be helpful here as we see this weak seasonal behaviour. We therefore include it.

The second part of your comment aims at the uncertainty existing in the photograph snow cover maps. You are right, shadows have been a problem when using RGB photography. Therefore, we tackled this issue by developing our method for shadow-affected 8-bit photographs, presented in Härer et al. (2016). We have shown in Härer et al. (2016) that the classifications using this approach has the same quality as the classification using the standard method for sunny photographs without shadows. And we carefully checked all images to ensure that the quality is as high as possible for each photograph. We mention here the highly conservative estimate of 5%. We checked each camera image visually and the value of 5% is the absolute maximum of error that we could think of in one of our classified images (Chapt. 4 in Härer et al., 2013). Usually, the classification error is below 1% if no major classification errors are obvious and thus not an issue. However, we will add these statements for clarification.

**Manuscript changes to 4)**

Removed p.10, l1 to l3: "This temporal development is potentially related to the sun angle, snow age, grain size or albedo development or other effects. A detailed investigation of the reasons of this effect is beyond this study but will be subject of future studies."
p.10, l6 to 12: "These results are promising and it needs to be investigated if the seasonal behaviour of the calibrated $NDSI$ thresholds can be attributed to the elevation and azimuth angles of the sun. The correlation r between azimuth angle and $NDSI$ is 0.75 for RCZ and 0.42 for VF. For sun elevation, r is 0.77 for RCZ and 0.54 for VF. The sun angles thus are correlated to the seasonal development but do not fully explain the behaviour. The temporal development is thus potentially also related to snow age, grain size, albedo development or other effects. These might also explain the observed variability

within the seasons. A detailed investigation of this variability is however beyond this study but will be subject of future studies."

p.4, l4 to 5: "The photographs are recorded as 8-bit data with three colour channels (red, green and blue; RGB) on an hourly basis for RCZ and three times a day for VF."

p.6, l12 to 18: "The photograph snow cover maps did have even in the case that an insufficient classification algorithm was used for a specific situation less than 5% misclassified pixels in the worst case region of the photograph in Chapt. 4 in Härer et al. (2013). It was also shown that the classification of shadow-affected photographs are of the same quality as sunny photographs (Chapt. 4 in Härer et al., 2016). As for this study, every classified image was visually inspected and no major snow classification errors comparable to our worst case example in the previous publication were found, we expect a relative misclassification error of 1%."

**Specific comments:**

Note: We do not show each of the manuscript changes for the specific comments here as the changes are obvious by the answer. Nonetheless, the changes will be denoted in the new manuscript.

**P1L12:** Earth not earth

**Answer:** Thank you for the correction, we will revise it.

**P3L22:** glacierized not glaciered

**Answer:** You are right, we will change it.

**P3L23-25:** It could be useful to show the spectral profiles of the snow-free substratum (limestone is more reflective in the visible range than gneiss).

**Answer:** We think that adding a figure is a bit too much here as only two bands are interesting for the NDSI, however we will add a paragraph describing the general spectral behaviour of limestone and gneiss with respect to the NDSI calculation in the results and discussion section to explain the different mean NDSI values.

**P4L13:** I do not think that this statement is true "no atmospheric correction is applied (..) the majority of studies that apply the NDSI for snow cover mapping". Many studies use the MOD10 snow products, or TMSCAG for Landsat, which use surface reflectances.

**Answer:** We will rephrase this sentence as there are also many studies that use atmospheric correction. We will clarify this.

**P3L17:** NDSI and NDSI_thr are written in equation mode, sometimes in plain text.

**Answer:** Thank you, we will change it.

**P4L9:** photographs not photographies. What is the acquisition time of the camera? is it synchronous to Landsat overpass time?

**Answer:** The photographs at RCZ are taken on an hourly basis and at VF in the morning, at noon and in the afternoon and thus the Landsat image is calibrated with a photograph that is recorded within the same hour at RCZ and within three hours at VF. We will add a statement for clarification.

**P4L31:** did you find a difference in the results between Landsat 8 and the Landsat 5/7? Landsat 8 instrument has higher radiometric resolution which improves snow classification in mountains (less saturation, higher SNR in shaded slopes).

**Answer:** We also had this thought at the beginning. However, we would need more acquisitions for this investigation. And the strong variability as described in Table 1 and Figure 7b superimposed on the seasonal threshold behaviour probably hides the signal between different sensor systems.

**P6L8:** Note that this metric is usually referred to as accuracy and may not be a robust performance measure when the number of a class is much greater than the number of the other one.

**Answer:** Thank you, we will use the term 'accuracy'. In general, each performance measure has a weakness. In our case however, both investigated catchments are partially glacierized. A minimum area of snow or ice is thus left in summer. It thus should not be a major problem in our catchments. Moreover, a prerequisite for our calibration method is that there are snow covered areas as well as areas free of snow in the photograph and thus also in the Landsat scene (p.4, l.13).

**P7L29:** if vertical, these rock faces are not visible in images captured by nadir-looking sensors like Landsat 7.

**Answer:** Thank you for bringing up this mistake. We are investigating summer dates here. Low and flat areas are also snow-free in this time of the year, the sentence was deleted.

**P7L31:** "NDSI reflectances" does not make sense

**Answer:** You are right, we will correct this in the complete manuscript.

**P8L10-12:** this sentence is not clear to me.

**Answer:** We will clarify the sentence. The 'uncertainty' term is maybe again inexact. The percentages outline the differences in snow cover between the standard and the calibrated threshold.

**P8L31:** fitted against what? Day of year I think.

**Answer:** Yes, this is true. We will clarify this.

**P9L18:** why not using real MODIS images instead?

**Answer:** We focus on Landsat and the scale of 30m in our study. The aggregation up to 990m is only an experimental setup to outline if the calibration of NDSI thresholds is needed for larger pixel sizes than 30m and we want to analyse different scales here. It is thus not the objective to evaluate a single snow cover product like MOD10/MYD10 which in addition is atmospherically corrected and does not use the 0.4 threshold anymore (see your general comment 1). Moreover, our small catchments are not the best experimental setup for MODIS.

**P9L25:** if I understand well, the resampling to 500m has increased the optimal threshold value. Can you think of an explanation?

**Answer:** Sorry, there is a misunderstanding. We simply say here that the NDSI threshold of 0.4 seems to be a really good estimate at a pixel size of 500m. And the NDSI threshold increases for RCZ and decreases for VF so we do not have any trend here. We add a sentence to underline this.

**Fig. 5:** rainbow colormaps are not recommanded (Borland 2007). I am also surprised by the choice of the projection (plate carrée? non-equidistant projections are not recommended for this kind of maps)

**Answer:** These figures are equidistant. It is the standard Matlab output using latitude and longitude in m. We will clarify this by adding the unit and we will also follow your suggestion to adapt the colormap.

[revised manuscript text omitted]
  column̲s̲ ̲f̲r̲o̲m̲ ̲l̲e̲f̲t̲ ̲t̲o̲ ̲r̲i̲g̲h̲t̲ ̲a̲r̲e̲  the camera calibrated SCA,  the standard threshold SCA , and  their differences at VF . The different rows show different scaling factors, b̲e̲i̲n̲g̲ 1 (30 m)  3 (90 m), 7 (210 m), 17 (510m) and  33 (990 m) f̲r̲o̲m̲ ̲t̲h̲e̲ ̲t̲o̲p̲ ̲t̲o̲ ̲t̲h̲e̲ ̲b̲o̲t̲t̲o̲m̲. The concurrent photograph in **b)** depicts the snow situation at VF in our example. The analysis of all investigation dates in **c)** shows a̲t̲ ̲w̲h̲i̲c̲h̲ ̲p̲i̲x̲e̲l̲ ̲s̲i̲z̲e̲ h̲o̲w̲ ̲m̲a̲n̲y̲ ̲o̲f̲ ̲t̲h̲e̲ camera calibrated and standard threshold snow cover maps become  identical . The s̲p̲a̲t̲i̲a̲l̲ ̲r̲e̲s̲o̲l̲u̲t̲i̲o̲n̲s̲ ̲o̲f̲ ̲t̲h̲e̲ ̲S̲e̲n̲t̲i̲n̲e̲l̲-̲ ̲2̲,̲ ̲L̲a̲n̲d̲s̲a̲t̲,̲ ̲M̲O̲D̲I̲S̲ ̲a̲n̲d̲ ̲N̲O̲A̲A̲ ̲A̲V̲H̲R̲R̲ ̲s̲a̲t̲e̲l̲l̲i̲t̲e̲s̲ a̲r̲e̲ ̲o̲u̲t̲l̲i̲n̲e̲d̲ ̲f̲o̲r̲ ̲o̲r̲i̲e̲n̲t̲a̲t̲i̲o̲n̲.